# A new molecular classification to drive precision treatment strategies in primary Sjögren's syndrome

Perrine Soret[1,29], Christelle Le Dantec[2,29], Emiko Desvaux[1,2], Nathan Foulquier[2], Bastien Chassagnol [1], Sandra Hubert[1], Christophe Jamin [2,3], Guillermo Barturen [4], Guillaume Desachy[1], Valérie Devauchelle-Pensec[2,3], Cheïma Boudjeniba[1], Divi Cornec [2,3], Alain Saraux[2,3], Sandrine Jousse-Joulin[2,3], Nuria Barbarroja[5], Ignasi Rodríguez-Pintó [6], Ellen De Langhe [7], Lorenzo Beretta[8], Carlo Chizzolini[9], László Kovács[10], Torsten Witte[11], PRECISESADS Clinical Consortium*, PRECISESADS Flow Cytometry Consortium*, Eléonore Bettacchioli[3], Anne Buttgereit[12], Zuzanna Makowska[12], Ralf Lesche[12], Maria Orietta Borghi[13], Javier Martin[14], Sophie Courtade-Gaiani [1], Laura Xuereb[1], Mickaël Guedj[1], Philippe Moingeon [1], Marta E. Alarcón-Riquelme [4], Laurence Laigle[1] & Jacques-Olivier Pers [2,3✉]

There is currently no approved treatment for primary Sjögren's syndrome, a disease that primarily affects adult women. The difficulty in developing effective therapies is -in part-because of the heterogeneity in the clinical manifestation and pathophysiology of the disease. Finding common molecular signatures among patient subgroups could improve our understanding of disease etiology, and facilitate the development of targeted therapeutics. Here, we report, in a cross-sectional cohort, a molecular classification scheme for Sjögren's syndrome patients based on the multi-omic profiling of whole blood samples from a European cohort of over 300 patients, and a similar number of age and gender-matched healthy volunteers. Using transcriptomic, genomic, epigenetic, cytokine expression and flow cytometry data, combined with clinical parameters, we identify four groups of patients with distinct patterns of immune dysregulation. The biomarkers we identify can be used by machine learning classifiers to sort future patients into subgroups, allowing the re-evaluation of response to treatments in clinical trials.

[1] Institut de Recherches Internationales Servier, Departments of Translational Medicine and Immuno-Inflammatory Diseases Research and Development, Suresnes, France. [2] LBAI, UMR1227, Univ Brest, Inserm, Brest, France. [3] CHU de Brest, Brest, France. [4] Department of Medical Genomics, Center for Genomics and Oncological Research (GENYO), Granada, Spain. [5] Reina Sofia Hospital, Maimonides Institute for Research in Biomedicine of Cordoba (IMIBIC), University of Cordoba, Cordoba, Spain. [6] Hospital Clinic, Institut d'Investigacions Biomèdiques August Pi i Sunyer, Barcelona, Catalonia, Spain. [7] Skeletal Biology and Engineering Research Center, KU Leuven and Division of Rheumatology, UZ Leuven, Belgium. [8] Scleroderma Unit, Referral Center for Systemic Autoimmune Diseases, Fondazione IRCCS Ca'Granda Ospedale Maggiore Policlinico di Milano, Milan, Italy. [9] Immunology & Allergy, University Hospital and School of Medicine, Geneva, Switzerland. [10] University of Szeged, Szeged, Hungary. [11] Klinik für Immunologie und Rheumatologie, Medical University Hannover, Hannover, Germany. [12] Pharmaceuticals Division, Bayer Pharma Aktiengesellschaft, Berlin, Germany. [13] Università degli studi di Milano, Milan, Italy. [14] Institute of Parasitology and Biomedicine López-Neyra, Consejo Superior de Investigaciones Científicas (IPBLN-CSIC), Granada, Spain. [29] These authors contributed equally: Perrine Soret and Christelle Le Dantec. *Lists of authors and their affiliations appear at the end of the paper. ✉email: pers@univ-brest.fr

Primary Sjögren's syndrome (pSS) is a chronic, disabling, complex systemic autoimmune disease that mostly affects adult women and still lacks a specific therapy. Although the involvement of salivary and lachrymal glands is the hallmark of the disease, during pSS progression, various organs and systems can be involved including joints, lungs, kidneys, liver, nervous and musculoskeletal system[1]. Thus, the clinical spectrum of the disease ranges from a benign slowly progressive autoimmune exocrinopathy to a severe systemic disorder with significant symptom heterogeneity and scattered complications. The diagnosis of pSS is currently based upon a combination of clinical, serological, histological, and functional parameters which are most often only satisfied at a late stage of the disease, i.e., when glandular dysfunction and symptoms already severely affect a patient's overall quality of life. Moreover, one fifth of pSS patients may present major organ involvement with potentially severe end-organ damage[2] and five percent of patients may also develop non-Hodgkin's lymphoma[3]. Primary SS is one of the few prototypic diseases to link autoimmunity, cancer development and infections, offering unique insights in many areas of basic science and clinical medicine. However, the pathogenesis of the disease remains elusive. Specifically, limited knowledge of existing pSS disease variants arguably represents the greatest obstacle to improve patients' diagnosis and identify patients' subsets in view of early stratification and personalized treatment[4]. It was recently shown in the PRECISESADS IMI JU project that systemic autoimmune diseases exhibit a diverse spectrum and a complex nuanced or overlapping molecular phenotype with four clusters identified, representing 'inflammatory', 'lymphoid', 'interferon' and 'healthy-like' patterns each including all diagnoses and defined by genetic, clinical, serological and cellular features[5]. Many of them share susceptibility genes[6] and an overexpression of interferon (IFN) inducible genes known as the IFN signature is observed in many of these patients[7]. Such autoimmune diseases are driven by numerous environmental factors, therefore displaying a marked variability in their natural course as it relates to their initiation, propagation and flares.

The present study was undertaken to establish a precise molecular classification of patients affected by pSS into more homogeneous clusters whatever their disease phenotypes, activity or treatment. We report herein on the integrated molecular profiling of 304 pSS patients compared to 330 matched healthy volunteers (HV) performed using high-throughput multi-omics data collected within the PRECISESADS IMI JU project (genetic, epigenomic, transcriptomic, combined with flow cytometric data, multiplexed cytokines, as well as classical serology and clinical data). We identify 4 groups of patients with distinct patterns of immune dysregulation. The Cluster 1 (C1), C3 and C4 display a high IFN signature reflecting the pathological involvement of the IFN pathway, but with various Type I and II IFN gene enrichment. C1 has the strongest IFN signature with both Type I and Type II gene enrichment when compared to C3 (intermediate) and C4 (lower). C4 has a Type II gene enrichment stronger than Type I and equivalent to C3 while C3 has the opposite composition. C2 exhibits a weak Type I and Type II IFN signature with no other obvious distinguishable profile relative to HV. We further characterized C1, C3 and C4 using multi-omics and clinical data. C1 patients present a high prevalence of SNPs, C3 patients an involvement of B cell component more prominent than in the other clusters and especially an increased frequency of B cells in the blood while C4 patients have an inflammatory signature driven by monocytes and neutrophils, together with an aberrant methylation status. Algorithms derived from machine learning discriminate the 4 clusters based on distinct biomarkers that can be easily used in a composite model to stratify patients in clinical trials. This composite model is validated by using an independent inception cohort of 37 pSS patients. In conclusion, this work provides a clear understanding of pSS heterogeneity providing clinically and immunopathologically relevant signatures to guide precision medicine strategies. Decision trees coming from this patient classification have an immediate application to re-evaluate response to treatments in clinical trials.

## Results

**Four functional molecular clusters of pSS patients were identified.** Our initial study population comprised 382 pSS patients enrolled in the PRECISESADS cross-sectional study. Following complete quality control and diagnosis validation (each patient had to present either anti-SSA/Ro antibody positivity or focal lymphocytic sialadenitis with a focus score of $\geq 1$ foci/mm[2]), 78 patients were removed (Supplementary Fig. 1a–c). Patient characteristics are presented in Table 1. To perform the clustering of the remaining 304 samples, transcriptomics data were analyzed with a semi-supervised robust approach previously applied to breast cancer[8] that iterates unsupervised and supervised steps and relies on the concordance between 3 methods of clustering (see Methods). Samples were divided into a discovery set and an independent validation set, representing 75 and 25% of samples, respectively. The discovery set allowed to cluster patients in four groups, as confirmed in the validation set (Fig. 1a). When the two sets were merged, Cluster 1 (C1) contained 101 patients (33.2%), Cluster 2 (C2) 77 patients (25.3%), Cluster 3 (C3) 88 patients (28.9%) and Cluster 4 (C4) 38 patients (12.5%). The supervised step allowed to select a subset of 257 top genes discriminating the 4 clusters of patients (Supplementary Fig. 2) and divided into 3 modules: M.a (105 genes), M.b (20 genes) and M.c (132 genes). An enrichment analysis was used to annotate each gene module, showing that M.a was enriched in IFN signaling, M.b in lymphoid lineage pathways and M.c in inflammatory and myeloid lineage transcripts (Supplementary Fig. 3). C1, and to a lesser extent C3, presented overexpression of gene module M.a, whereas C3 showed overexpression of M.b as well and C4 strong overexpression of M.c (Fig. 1a). Because C2 had no obvious discernible pattern, healthy volunteers (HV) were assigned to the 4 molecular clusters distance to centroids (Fig. 1b). When projected into the patient population, HV did not constitute a separate cluster but mainly matched with C2 (0.5%, 93%, 4% and 2.5% of HV merged with C1, C2, C3, and C4, respectively). This means that the C2 transcriptional signature is not different from HV, at least at the blood level. Interestingly, our data are consistent with the previous observation of a healthy-like patient group detected in a pooled population of 7 different autoimmune diseases[5].

We then assessed whether covariates like systemic treatments could drive the transcriptome-based clustering. Indeed, half of the pSS patients were treated with either anti-malarials, immunosuppressants, or steroids at the time of the visit with a statistically significant difference in the distribution among the four clusters ($p$-values were respectively 0.002 for anti-malarials, <0.001 for immunosuppressants and steroids) (Table 2). When compared to the 3 other clusters, a higher proportion of patients treated with anti-malarials in C2 and a higher proportion of patients receiving immunosuppressants or steroids in C4 were observed. Importantly, sensitivity analyses of treated versus untreated patients in each cluster showed no impact of treatments on cluster distribution (Supplementary Fig. 4).

**In depth functional pathway analysis of individual pSS clusters.** To investigate molecular processes and their biological function underlying each of the pSS patients' clusters, specific differentially expressed genes (DEG) signatures compared to HV were assessed using Limma in the 4 clusters. Ingenuity Pathway Analysis (IPA)

**Table 1 Healthy volunteers (HV) and Primary Sjögren's syndrome (pSS) patient characteristics.**

| | | | HV (N = 330) | pSS Discovery (N = 227) | pSS Validation (N = 77) | pSS All (N = 304) |
|---|---|---|---|---|---|---|
| Demography | | | | | | |
| Age | | n | 330 | 227 | 77 | 304 |
| | | Mean ± SD | 53.294 ± 10.998 | 58.524 ± 13.440 | 58.039 ± 13.554 | 58.401 ± 13.448 |
| Gender | | n | 330 | 227 | 77 | 304 |
| | Female | n (%) | 302 (91.52) | 211 (92.95) | 71 (92.21) | 282 (92.76) |
| Obesity (BMI > = 30) | | n | 328 | 218 | 74 | 292 |
| | Yes | n (%) | 24 (7.27) | 30 (13.76) | 3 (4.05) | 33 (11.30) |
| Race | | n | 330 | 227 | 77 | 304 |
| | Asian | n (%) | 2 (0.61) | 1 (0.44) | 1 (1.30) | 2 (0.66) |
| | Black/African American | n (%) | — | — | 1 (1.30) | 1 (0.33) |
| | Caucasian/White | n (%) | 328 (99.39) | 224 (98.68) | 74 (96.10) | 298 (98.03) |
| | Other | n (%) | — | 2 (0.88) | 1 (1.30) | 3 (0.99) |
| Diagnostic criteria | | | | | | |
| Focus score > 1 | | n | — | 82 | 27 | 109 |
| | Yes | n (%) | — | 73 (89.02) | 24 (88.89) | 97 (88.99) |
| Anti-SSA positivity | | n | — | 227 | 77 | 304 |
| | Yes | n (%) | — | 205 (90.30) | 69 (89.61) | 274 (90.13) |
| Disease activity | | | | | | |
| Disease duration, years | | n | — | 225 | 77 | 302 |
| | | Mean ± SD | — | 10.788 ± 7.535 | 11.094 ± 9.620 | 10.866 ± 8.101 |
| Disease activity (PGA*) | | n | — | 211 | 75 | 286 |
| | | Mean ± SD | — | 25.687 ± 18.976 | 24.840 ± 20.984 | 25.465 ± 19.488 |
| ESSDAI (**) | | n | — | 133 | 60 | 193 |
| | | Mean ± SD | — | 4.609 ± 5.358 | 4.850 ± 5.495 | 4.684 ± 5.388 |
| ESSPRI (**) | | n | — | 106 | 44 | 150 |
| | | Mean ± SD | — | 5.176 ± 2.286 | 4.568 ± 2.648 | 4.998 ± 2.405 |

n: Number of patients with available information.
(*) PGA: Physician Global Assessment.
(**) collected in a substudy.

was subsequently applied to determine the most significantly dysregulated canonical pathways with Benjamini–Hochberg false discovery rate (FDR) adjusted *p*-value ≤ 0.05 and absolute fold change (FC) ≥ 1.5. As a result, 284 DEG were found significant in C1, 301 DEG in C3 and 1686 DEG in C4 (Supplementary Data 1).

Since no DEG were noticed in C2 when compared to HV, only C1, C3, and C4 were functionally annotated. Top 20 significant canonical pathways within each DEG signature are presented in Supplementary Data 2 and pathways related to the most significantly enriched immunological responses are reported as radar plots in Fig. 1c. While all 3 clusters were enriched in genes involved in antiviral and anti-bacterial responses indicative of an innate-mediated activation profile, C1 was mainly enriched with IFN-related pathways including IFN signaling, role of pattern recognition receptors for bacteria and viruses and Interferon Regulatory Factor (IRF) activation. Notably, C3 and C4 were further characterized by alterations in biological networks linked to adaptive immunity. Specifically, significant activation of canonical pathways related to B cell activation such as B cell receptor signaling, and B cell development were observed in C3. In addition, comparative analyses provided evidence for IL7-signaling up-regulation and LXR/RXR activation in C3 compared to C1.

Interestingly, C4 was the endotype with the highest number of DEG compared to HV with highly heterogeneous dysregulated canonical pathways. Ingenuity pathway analysis confirmed the activation of T and B lymphocyte related pathways reflecting Th1 and Th2 activation, B cell receptor signaling, together with prominent inflammatory signatures most particularly linked to cytokine signaling (IL-6 and IL-10 signaling, IL-15 production, STAT-3 pathway).

Further upstream regulator analysis predicted significant activation of IFN-α in all three clusters, as well as CpG ODN in C3 and LPS, IFNγ, TNF-α, and IL-4 in C4, further highlighting B cell activity and inflammatory responses in C3 and C4, respectively.

Noteworthy, while C2 displayed no DEG compared to HV, 14 genes were differentially expressed in C2 patients positive for SSA antibodies compared to HV whereas only 2 DEG were found in SSA-negative C2 patients. These SSA-positive C2 patients were characterized by significant enrichment in IFN-related genes compared to HV including *IFI44, IFI44L, IFI6, IFIFT1, IFIT3, ISG15, MX1, OAS3, SERPING1,* and *SIGLEC1* (Supplementary data 1).

To further characterize patient cluster variability at a molecular level, we then used the blood transcriptome modular repertoire recently established on an expended range of disease and pathological states. The latter includes 382 transcriptome modules based on genes co-expression patterns across 16 diseases and 985 unique transcriptome profiles[9]. Again, no aggregate was found differentially expressed in C2 confirming the healthy-like profile of these patients, whereas an up-regulated IFN signature dominated in C1, C3, and C4 (Fig. 2). In C4, the most induced modules include genes associated with inflammation and neutrophils. As the highest inflammatory phenotype, C4 is associated with a hypercytokinemia/hyperchemokinemia

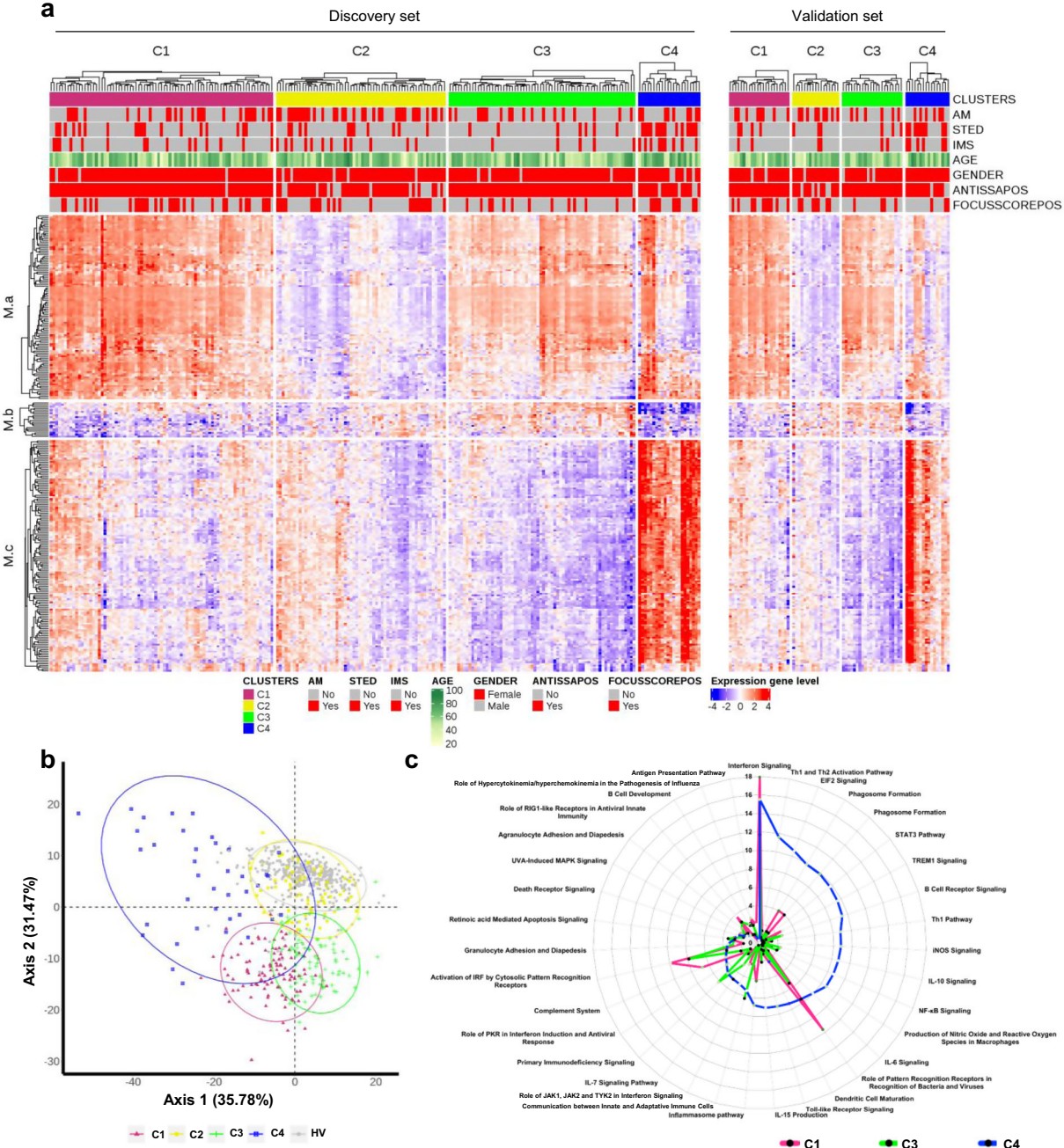

**Fig. 1 Molecular pattern distribution is represented by 4 clusters of pSS patients with different canonical pathways. a** Heatmap performed for 304 pSS patients (Discovery set: 227, Validation set: 77) showing the distribution of gene transcripts across the 4 clusters. In columns patients are grouped by cluster assignment and in rows genes are grouped by functional modules. Each subset of patients (discovery set on the left and validation set on the right) is presented separately. Red represents overexpression and blue represents under-expression. At the top of the figure annotations show: each of the treatment groups for each individual (AM: antimalarials, STED: steroids and IMS: immunosuppressors, red represents patients with treatment and gray represents patients without treatment), age (levels of yellow to green with yellow for younger patients and dark green for older patients), gender (red represents woman and gray represents man), ANTISSAPOS: anti-SSA/Ro antibody positivity, FOCUSSCOREPOS: focus score of ≥1 foci/mm$^2$ (red represents focus score of ≥1 foci/mm$^2$ and gray represents focus score of <1 foci/mm$^2$). **b** Scatterplot of the first two components PCA (performed for 304 pSS patient and 330 HV) model showing clearly defined clusters in signature gene. HV (gray dot) are confused with C2 cluster (yellow dot). **c** Top 20 most significant canonical pathways for each cluster. Radar plots are represented according to −log (p-value) (Fisher's exact test) associated to the most significant pathways of each cluster; C1 (pink), C3 (green), C4 (blue).

observed in modules (M13.16, M15.84, M16.80) consistent with an upregulation of the TNF-associated module (M16.47) and a downregulation of the TGFβ-associated module (M16.65) (Fig. 2). Some modules were under-expressed, such as those associated with both protein synthesis (M12.7, M11.1, M13.28, M14.80), B

cells (M13.27, M12.8) and T cells (M15.38, M14.42, M12.6). Genes mainly overexpressed in C1 were also implicated in inflammatory responses and neutrophils (A33, A35), in parallel with down-regulated B and T cell signatures (Supplementary Fig. 5). Moreover, distinct sub-modules expressed in opposite

**Table 2 Descriptive analysis of the clinical parameters by primary Sjögren's syndrome cluster.**

| | | C1 (n = 101) | C2 (n = 77) | C3 (n = 88) | C4 (n = 38) | p-value |
|---|---|---|---|---|---|---|
| Age, years | n | 101 | 77 | 88 | 38 | |
| | Mean ± SD | 57.327 ± 13.705 | 58.805 ± 13.688 | 57.250 ± 12.032 | 63.105 ± 14.790 | 0.10 |
| Gender | n | 101 | 77 | 88 | 38 | |
| Female | n (%) | 96 (95.05) | 71 (92.21) | 81 (92.05) | 34 (89.47) | 0.70 |
| Age at onset, years | n | 101 | 76 | 88 | 37 | |
| | Mean ± SD | 45.663 ± 14.475 | 50.428 ± 14.532 | 47.606 ± 12.687 | 51.739 ± 16.053 | 0.071 |
| Disease duration, years | n | 101 | 76 | 88 | 37 | |
| | Mean ± SD | 12.247 ± 8.921 | 8.965 ± 7.336 | 10.183 ± 7.210 | 12.625 ± 8.524 | 0.029 |
| Disease activity (PGA*) | n | 94 | 71 | 85 | 36 | |
| | Mean ± SD | 27.245 ± 20.535 | 22.718 ± 17.698 | 23.212 ± 18.766 | 31.556 ± 20.646 | 0.092 |
| ESSDAI | n | 70 | 52 | 44 | 27 | |
| | Mean ± SD | 5.029 ± 5.959 | 3.731 ± 4.594 | 4.227 ± 4.017 | 6.370 ± 6.828 | 0.10 |
| ESSPRI | n | 56 | 43 | 30 | 21 | |
| | Mean ± SD | 4.833 ± 2.460 | 5.031 ± 2.429 | 5.300 ± 2.703 | 4.937 ± 1.803 | 0.87 |
| Arthritis | n | 98 | 77 | 86 | 38 | |
| Past | n (%) | 39 (39.80) | 18 (23.38) | 20 (23.26) | 12 (31.58) | 0.016 |
| Present | n (%) | 2 (2.04) | 3 (3.90) | 4 (4.65) | 5 (13.16) | |
| Focus score > 1 | n | 96 | 29 | 21 | 14 | |
| Yes | n (%) | 39 (40.63) | 28 (96.55) | 17 (80.95) | 12 (85.71) | 0.4 |
| Anti-SSA positivity | n | 101 | 77 | 88 | 38 | |
| Yes | n (%) | 99 (99.00) | 56 (72.72) | 87 (98.86) | 31 (81.57) | <0.001 |
| Anti-SSB positivity | n | 100 | 77 | 86 | 38 | |
| Yes | n (%) | 61 (61.00) | 12 (15.58) | 39 (45.35) | 11 (28.95) | <0.001 |
| Hypergammabulinemia | n | 97 | 73 | 86 | 38 | |
| Past | n (%) | 23 (23.71) | 8 (10.96) | 9 (10.47) | 3 (7.89) | <0.001 |
| Present | n (%) | 44 (45.36) | 10 (13.70) | 41 (47.67) | 7 (18.42) | |
| Abnormal inflammatory indexes | n | 100 | 77 | 87 | 38 | |
| Past | n (%) | 28 (28.00) | 13 (16.88) | 20 (22.99) | 12 (31.58) | 0.003 |
| Present | n (%) | 35 (35.00) | 11 (14.29) | 22 (25.29) | 10 (26.32) | |
| Reduced C3 levels | n | 93 | 74 | 82 | 35 | |
| Past | n (%) | 13 (13.98) | 5 (6.76) | 11 (13.41) | 4 (11.43) | 0.8 |
| Present | n (%) | 7 (7.53) | 4 (5.41) | 5 (6.10) | 3 (8.57) | |
| Reduced C4 levels | n | 93 | 74 | 82 | 35 | |
| Past | n (%) | 13 (13.98) | 3 (4.05) | 9 (10.98) | 4 (11.43) | 0.10 |
| Present | n (%) | 10 (10.75) | 3 (4.05) | 3 (3.66) | 4 (11.43) | |
| Abnormal Creatinine | n | 98 | 77 | 88 | 38 | |
| Past | n (%) | 10 (10.20) | 4 (5.19) | - | 2 (5.26) | 0.009 |
| Present | n (%) | 5 (5.10) | 2 (2.60) | 7 (7.95) | 6 (15.79) | |
| Proteinuria | n | 65 | 58 | 56 | 25 | |
| Moderate | n (%) | 5 (7.69) | 2 (3.45) | 1 (1.79) | 3 (12.00) | 0.093 |
| Past | n (%) | 5 (7.69) | — | 3 (5.36) | — | |
| Current use of antimalarials | n | 101 | 77 | 88 | 38 | |
| Yes | n (%) | 33 (32.67) | 42 (54.55) | 24 (27.27) | 15 (39.47) | 0.002 |
| Current use of Immunosuppressants | n | 101 | 77 | 88 | 38 | |
| Yes | n (%) | 17 (16.83) | 14 (18.18) | 7 (7.95) | 15 (39.47) | <0.001 |
| Current use of steroids | n | 101 | 77 | 88 | 38 | |
| Yes | n (%) | 23 (22.77) | 14 (18.18) | 10 (11.36) | 23 (60.53) | <0.001 |

n: Number of patients with available information, (*) PGA: Physician Global Assessment.
Statistical tests performed: chi-square test of independence for categorial variable and Kruskal–Wallis test for continue variable.

directions allows to functionally discriminate C1 and C3. Patients from C3 demonstrated a significant under-expression of modules related to erythrocytes (A37; M9.2, M11.3) and cytokines/chemokines (A35; M15.84, M13.16) and an increased expression in some of the B cell modules (A1; M12.8) (Supplementary Fig. 5 and Fig. 2).

**IFN signatures**. Consistent with the literature, the most significantly enriched pathway confirmed to be up-regulated in all three clusters was the IFN signaling pathway (Fig. 2, Supplementary Fig. 5). In SLE, Chiche et al. have previously identified three strongly up-regulated IFN-annotated modules (M1.2, M3.4, and M5.12) from peripheral blood transcriptomic data, with for each module a distinct activation threshold[10]. Genes within the

M1.2 module are induced by IFNα, while other genes from both M1.2 and M3.4 are up-regulated by IFNβ, corresponding to a type I IFN signature. The M5.12 genes are poorly induced by IFNα and IFNβ alone but are rather up-regulated by IFNγ characterizing a type II IFN signature[11]. Moreover, transcripts belonging to M3.4 and M5.12 were only fully induced by a combination of Type I and Type II IFNs. Kirou et al. made similar observations and identified genes preferentially induced by IFNα or IFNγ[12]. The different z-scores were then calculated accordingly to characterize further the IFN signature observed in the various clusters (Fig. 3). All IFN z-scores were increased to some extent in C2 when compared to HV. In line with the strong signal observed, C1 patients had the highest Type I and type II scores. Interestingly, C3 had higher Type I IFN score than C4 but these 2 clusters were not different for Type II IFN score.

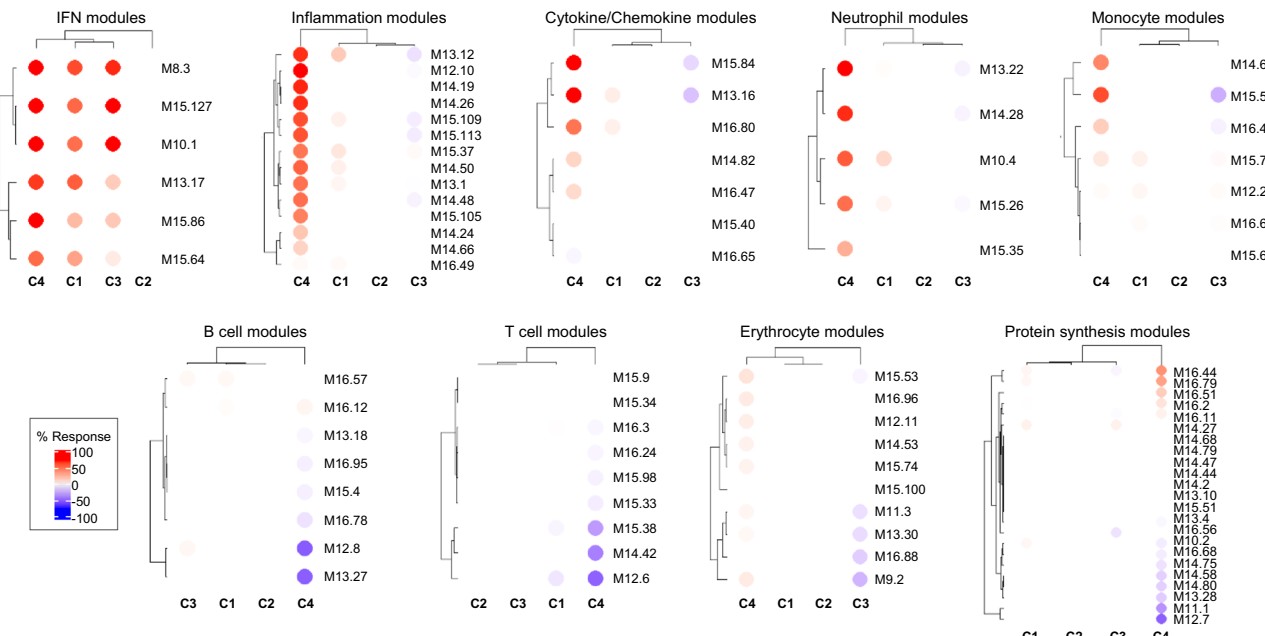

**Fig. 2 Patterns of abundance of the different modules distinguish the four pSS clusters.** Each heatmap, achieved with BloodGen3Module R package[9], represents one of the most significant patterns differentiating the four clusters of 304 pSS patients (C1: 101, C2: 77, C3: 88, and C4: 38) compared to 330 healthy volunteers (HV). These patterns correspond to modules associated with IFN, neutrophils, inflammation, cytokines/chemokines, protein synthesis, erythrocytes, monocytes, B cells and T cells. Columns on this heatmap corresponds to clusters. Each row corresponds to one of the modules associated with the pattern. For each module, the percentage of increased genes (from 0 to 100) and decreased genes (from 0 to 100) were calculated. A red spot on the heatmap indicates an increase in abundance of transcripts comprising a given module for a given cluster. A blue spot indicates a decrease in abundance of transcripts. The absence of color indicates no changes.

Upstream analysis of C4 DEG predicted IFNγ as an important regulator suggesting that Type II IFN activation was prominent in C4.

**Genome-wide association study analysis.** We investigated whether clusters showed any differences in the genetic contribution of risk alleles known to be associated with pSS[13–15]. Even in the mid-size cohort of patients analyzed (304 pSS and 330 HV), we unambiguously detected (with signals genome wide significance level $< 5 \times 10^{-8}$) 35 single nucleotide polymorphisms (SNPs) in C1 compared to only six in C3 and one in C4 (Fig. 4a, Supplementary Data 3). Interestingly, no significant enrichment was found in C2. The 35 SNPs assessed in C1 are found within genes associated with either the immune system (*HLA-DQB1, HLA-DQA1, HLA-DRA, HLA-C, HLA-G*), signal transduction (*NOTCH4*), developmental biology (*POU5F1*), gene expression (*DDX39B*) or cell cycle (*TUBB*). The presence of such significant genetic associations was already found in clusters of systemic autoimmune disease patients whose molecular disease pathway is the Type I IFN pathway[5]. Moreover, a strong association of SNPs with HLA class II genes was reported in SLE patients with a high level of autoantibodies[16]. One SNP (rs2734583) was common to C1 and C3 and is associated to the *DDX39* gene. Of note, DDX39B, the protein encoded by this gene, is required for the prevention of dsRNA formation during influenza A virus infection, thereby preventing the activation of the Type I IFN system[17]. The five others SNPs in C3 are nearby *HLA-DQA*, *HLA-DRA* (2 SNPs), *BTNL2* and *HCG23*. The only SNP (rs2247056) found in C4, also common with C1, is located in intron 1 of the *LINC02571* gene and was previously associated with a risk for developing SLE.

Linkage disequilibrium is a non-random association of alleles at different loci in a given population. When analyzing linkage disequilibrium (Fig. 4b) in the loci of the 35 SNPs detected in C1

and located on chromosome 6 (from base 29809362 to 32681631), three SNPs were strongly associated in *HLA-DQA1* locus (rs9272219, rs9271588, rs642093), five SNPs in *HLA-DRA | HLA-DQA1* locus (rs7195, rs1041885, rs3129890, rs9269043, rs7749057) and three SNPs in *HCG27 | HLA-C* locus (rs3130473, rs2394895 and rs3130467). Two other regions contain strongly associated SNPs. The *NOTCH4 | C6orf10* locus presented 5 associated SNPs (rs3130347, rs204991, rs3132935, rs7751896, rs9268220) as well as the *IER3 | DDR1* locus (rs3094122, rs6911628, rs3094112, rs2517576, rs3095151).

**Methylation analysis.** The methylation analysis was performed with a Benjamini Hochberg FDR < 0.1 and absolute ΔBeta > 0.075. Only two differentially methylated positions (DMPs) corresponding to two genes were found in C2. Those DMPs were common with the 3 other clusters (Fig. 5a) and were located in the TSS1500 shore of the *NLRC5* gene and in the 5'UTR of the gene encoding *MX1*, two genes involved in the IFN signature. NLRC5 plays a role in cytokine response and antiviral immunity through inhibition of NF-kappa-B activation and negative regulation of Type I IFN signaling pathways[18]. *MX1* encodes an IFN induced dynamic-like GTPase with antiviral activity which was proposed as a clinically applicable biomarker for identifying systemic Type I IFN in pSS[19].

145 DMPs corresponding to 87 genes and 96 DMPs corresponding to 56 genes were found in C1 and C3 respectively, whereas an aberrant methylation status with 8,445 DMPs corresponding to 3,636 genes characterized C4 (Fig. 5a). In order to test whether the methylation defect in C4 was associated with steroids treatment, we compared the 9 untreated to the 17 treated patients. No CpG with a Benjamini-Hochberg FDR adjusted *p*-value < 0.1 was found to be differentially methylated in treated versus untreated patients. A global hypomethylation of CpG was observed for all clusters (89.6% in C1, 100% in C2, 67.7% in C3

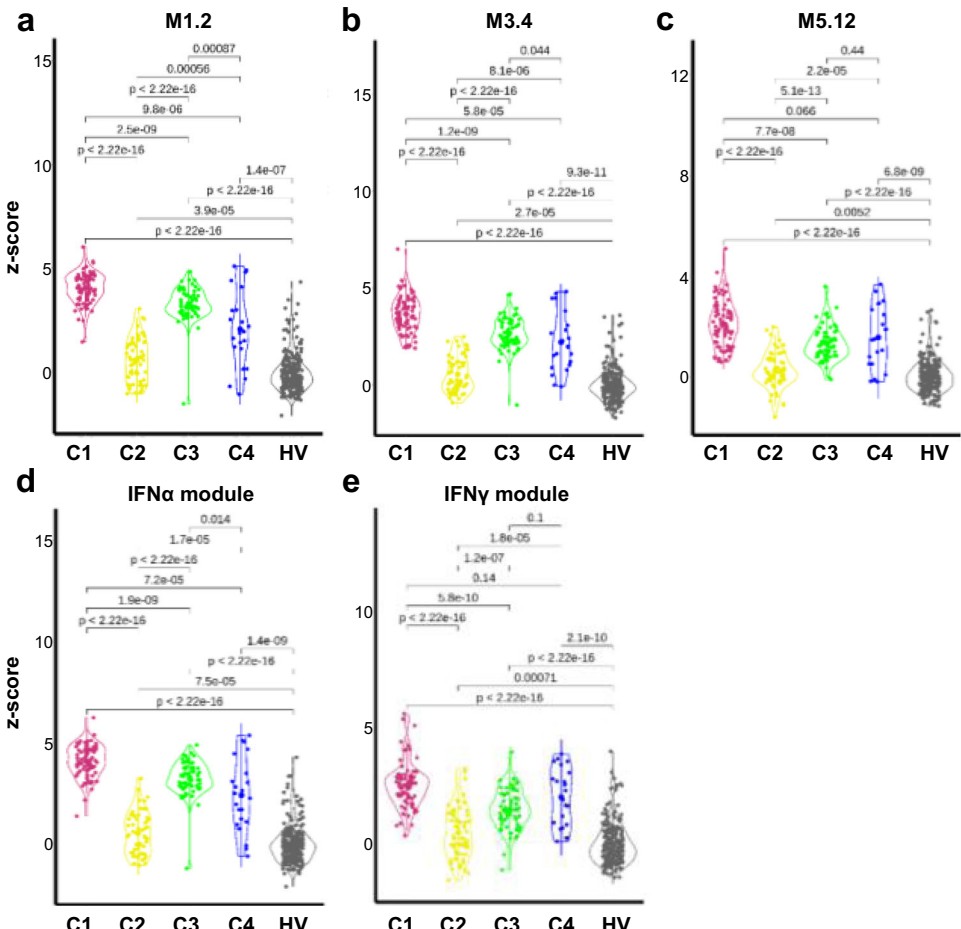

**Fig. 3 The 4 pSS clusters show typical IFN signature according to modular IFN z-scores.** IFN score analyses were performed for 304 pSS patients and 330 healthy volunteers (HV). Repartition of samples from the 4 pSS clusters are shown according to the most characterized IFN module z-scores. The genes (*IFI44, IFI44L, IFIT1* and *MX1*) of the M1.2 module (**a**) are induced by IFNα, while genes from both M1.2 and M3.4 (**b**) (*ZBP1, IFIH1, EIF2AK2, PARP9* and *GBP4*) are up-regulated by IFNβ. **c** The genes (*PSMB9, NCOA7, TAP1, ISG20* and *SP140*) from the M5.12 module are poorly induced by IFNα and IFNβ alone while they are up-regulated by IFNγ. Moreover, transcripts belonging to M3.4 and M5.12 were only fully induced by a combination of Type I and Type II IFNs[10]. Other modules identified genes preferentially induced by IFNα (*IFIT1, IFI44* and *EIF2AK2*) (**d**) or IFNγ (*IRF1, GBP1* and *SERPING1*) (**e**)[12]. Two-tailed pairwise Wilcoxon-rank sum test results are shown. Plots show median with error bars indicating ± interquartile range.

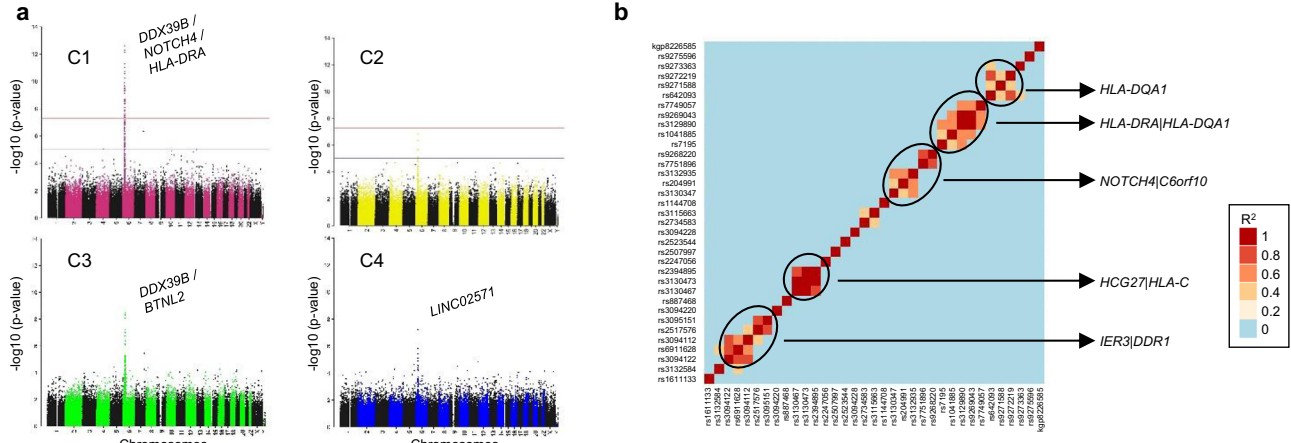

**Fig. 4 Cluster genome-wide association analyses (GWAS).** GWAS analysis was performed using Plink, an open-source whole genome association analysis toolset, using a logistical regression for 304 pSS (C1: 101, C2: 77, C3: 88 and C4: 38) patients and 330 healthy volunteers (HV) and each cluster was compared to HV. **a** Manhattan plots for each cluster are shown. **b** Linkage disequilibrium analysis in the loci of the 35 SNPs detected in C1 and located on chromosome 6 from base 29809362 to base 32681631. The $R^2$ correlation coefficient and linkage disequilibrium heatmap were obtained with Plink, and oncofunco R package, respectively. Strongest associations between SNPs are annotated.

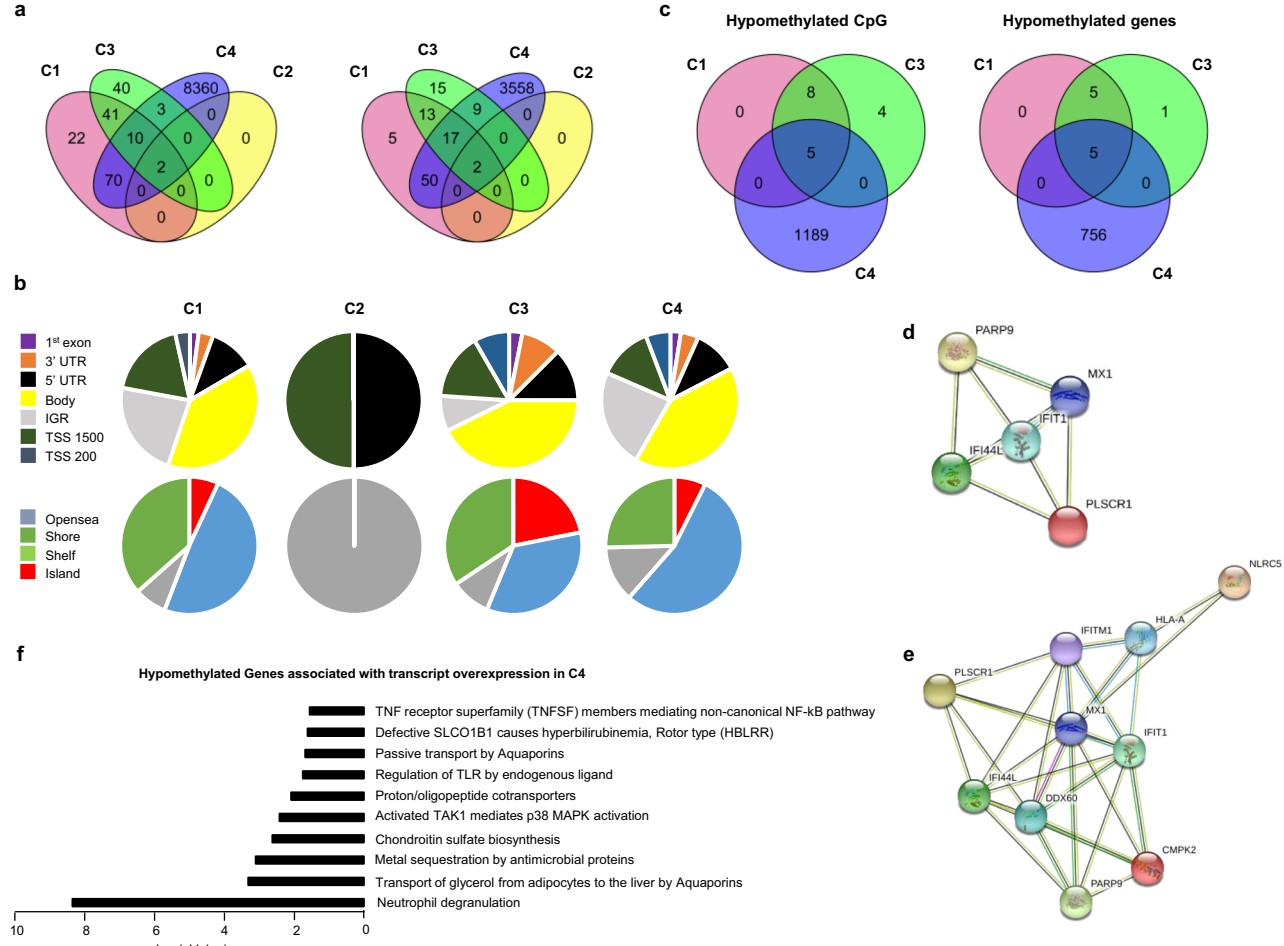

**Fig. 5 Methylation analysis confirms the strong IFN signature in C1 and C3 and reveals an aberrant methylation status in C4.** Whole blood methylation analysis was performed for 226 pSS patients (C1: 81, C2: 57, C3: 62, and C4: 26) and 175 healthy volunteers (HV) doing pairwise comparisons between each cluster and HV. **a** Venn diagram showing the overlap of differentially methylated CpG sites and genes between the 4 clusters with absolute ΔBeta > 0.075. **b** DMP distribution across the different genomic regions (gene body, 3'UTR, intergenic (IGR), 5'UTR, Exon 1, TSS 1500 and TSS 200; and according to the CpG density to CpG island, shelf, shore, and open sea. **c** Venn diagram showing the overlap of hypomethylated CpG and genes with absolute ΔBeta > 0.15 between the three IFN clusters. **d** Interaction network of these 5 genes common to the three clusters by STRING analysis with a confidence cut-off of 0.4 reveals a common IFN signature. **e** Interaction network of the 10 genes hypomethylated common to C1 and C3 by STRING analysis with a confidence cut-off of 0.4. **f** Reactome analysis[22] of the functional pathways enriched for the 126 genes hypomethylated and over expressed in C4 (absolute ΔBeta > 0.15, FC ≥ 1.5).

and 80.4% in C4). Because functionally important DNA methylation occurs in promoter regions and in CpG islands[20], DMP distribution across the different genomic regions was investigated (Fig. 5b). A higher representation of DMPs in the promoter region was found in C3 (36.4%) and C1 (33.1%) when compared to C4 (29.1%). The consequence was a lower representation of DMPs in intergenic regions for C3 (8.8%) compared to C1 (22.8%) and C4 (23.1%). To gain insight on this pattern, we divided the probes according to CpG islands; shores (regions up to 2 kb from CpG island), shelves (regions from 2 to 4 kb from CpG island) and open sea (the rest of the genome). Interestingly, 21.8% of the DMPs for C3 were located in CpG islands versus 6.9 and 7.4% for C1 and C4, respectively.

To identify the most robust and significant signature of hypo- and hyper-methylated genes, we fixed the ΔBeta cut-off at 0.15. Regarding hypomethylated CpGs, 13 DMPs were found in C1, 17 in C3 and 1,194 in C4, corresponding to 10, 11 and 761 hypomethylated genes, respectively. Five genes with hypomethylated DMPs were common to these 3 clusters (*IFI44L*, *IFIT1*, *MX1*, *PARP9* and *PLSCR1*) (Fig. 5c), corresponding to genes

reported to present strong interactions (Fig. 5d). Interestingly, these genes were also significantly hypomethylated in C2 when compared to HV (Supplementary Fig. 6). Of note, 5 additional genes (*HLA-A*, *DDX60*, *CMPK2*, *IFITM1* and *NLRC5*) were common to C1 and C3 and were also strongly associated with the previous ones, reinforcing the IFN signature in these two clusters (Fig. 5e). These common 10 hypomethylated genes are implicated in defense responses to virus and are induced by IFN[21].

The remaining 756 hypomethylated genes in C4 were mainly associated with the neutrophil degranulation pathway. Regarding hypermethylated CpGs, 41 DMPs corresponding to 25 genes were only found in C4. Those genes are mainly implicated in translocation of ZAP-70 to the immunological synapse, phosphorylation of CD3 chains including zeta, platelet activation, signaling and aggregation, homeostasis and PD-1 signaling.

Combining transcriptomic (FC ≥ 1.5) and methylomic (absolute ΔBeta > 0.15) analyses, the transcripts of 8, 8 and 126 genes were found to be increased in association with a decreased methylation status in C1, C3 and C4, respectively. Interestingly, the previously isolated 5 common hypomethylated genes

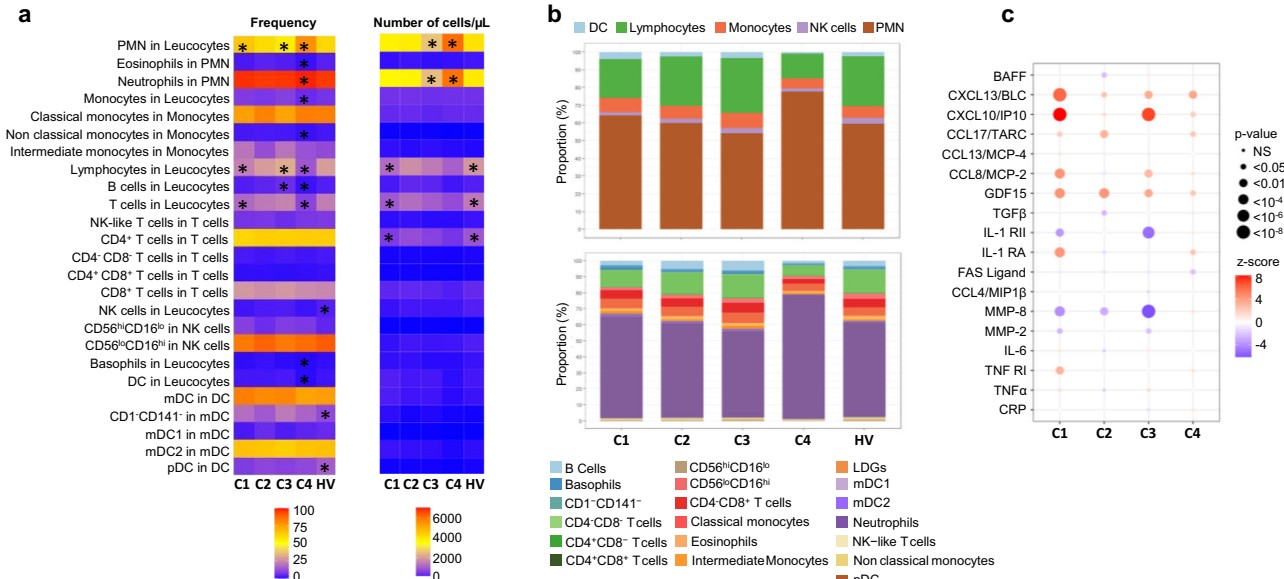

**Fig. 6 Cell subset distribution in blood and cytokines, chemokines and inflammatory mediators in serum in the 4 clusters and healthy volunteers (HV).**
**a** Flow cytometry analysis was performed for 283 patients (C1: 96, C2: 71, C3: 80, and C4: 36) and 309 HV. The 2 heatmaps show the mean distribution of blood cell subsets in frequency (0–100%) and in absolute numbers (per µL of blood) across the 4 clusters and HV assessed by flow cytometry. Columns represent clusters and HV and rows the different cell subsets. The asterisk means that the cluster (or HV) is statistically different from all the others.
**b** Flow cytometry data represented by bar charts cell types proportion per cluster. **c** Serum mediators were analyzed for 192 pSS patients (C1: 67, C2:48, C3: 61, C4:16) and 171 HV. Patient and HV distribution according to each analyzed variable is described in Methods. CXCL13/BLC, FAS Ligand, GDF-15, CXCL10/IP-10, CCL8/MCP-2, CCL13/MCP-4, CCL4/MIP-1β, MMP-8, CCL17/TARC, IL-1 RII, TNF-RI, and IL1-RA were measured using the Luminex system and expressed as pg/ml. Soluble MMP-2, CRP, TNFα, IL-6, BAFF, and TGFβ were measured by the quantitative sandwich enzyme immunoassay technique and expressed as pg/ml. Cytokine or chemokine concentration levels for each cluster were compared to HV. Statistical significance is determined using a one-way ANOVA followed by post-hoc Tukey's test. The significance between the cluster and HV is represented as bullet ranging from small (non-significant) to big (significant). The direction of the association is shown as the z-score where red bullet is up-regulated, and blue bullet is down-regulated.

implicated in IFN signaling were also overexpressed at the transcriptional levels in the 3 clusters. Transcript overexpression was strongly associated with hypomethylation in C1 (8/10) and C3 (8/11) and to a less extend in C4 (126/761). Among the 126 genes from C4, 21 were implicated in neutrophil degranulation which constitutes the most relevant pathways according to Reactome Pathway Database[22] (Fig. 5f). Only 6/25 transcripts were repressed in association with an increased methylation status of their genes in this cluster (*CD247*, *CD3G*, *CDC25B*, *CXCR6*, *TBC1D4*, *UBASH3A*).

**Flow cytometry analysis.** As significant alterations in patterns of peripheral blood leukocytes have been previously described[23,24], we then investigated the composition of leukocyte subsets in the various clusters. (Fig. 6a, b, Supplementary Fig. 7). In C2, the frequency and absolute numbers were similar to HV in all the different subsets analyzed. An increase in the frequency of monocytes and lymphocytes characterized C3, in association with a marked increase in the frequency of B cells. At the same time, a lymphopenia affecting mainly T cells was found in C1. Finally, the most distinguishable cluster in terms of distribution and absolute number of cells is C4. Specifically, C4 was characterized by higher percentages and absolute numbers of PMN (especially neutrophils) in peripheral blood in comparison with those in other clusters and HV. Conversely, the percentages of lymphocytes (B and T cells) and monocytes were markedly decreased in C4 compared to either the controls or the other clusters. Finally, lower frequencies and absolute numbers of basophils and DCs were also found in this cluster.

An in-depth analysis of the different cell subpopulations was then conducted. First, monocytes represent a heterogeneous cell population in terms of both phenotype and function. Based on

the expression of CD14 and CD16, 3 monocyte subsets can be defined, including classical (CD14++CD16−), intermediate (CD14++CD16+) and non-classical (CD14+CD16++). Classical monocytes are critical for the initial inflammatory response, can differentiate into macrophages in tissue and contribute to chronic disease. Intermediate monocytes are highly phagocytic cells that produce high levels of ROS and inflammatory mediators. Non classical monocytes have been widely viewed as anti-inflammatory, as they maintain vascular homeostasis and constitute a first line of defense in recognition and clearance of pathogens[25]. Interestingly, the frequency and absolute number of intermediate monocytes were increased in C1 and C3 whereas the frequency of classical monocytes was decreased when compared to the 2 others and the nonclassical subset was markedly decreased in C4, in line with the inflammatory response observed in these different clusters.

Second, NK cells are defined by the expression of CD56 and the lack of CD3-TCR complex. Moreover, based on CD16 and CD56 expression levels, they are classified in two subsets: CD56hiCD16lo and CD56loCD16hi. The latter NK cell subset mediates natural and antibody-dependent cellular cytotoxicity, exhibiting high levels of perforin and enhanced killing. In contrast, CD56hiCD16lo NK cells are characterized by low levels of perforin, and are primarily specialized for cytokine production including IFN[26,27]. Accordingly, the frequency of CD56hiCD16lo NK cells subset over CD56loCD16hi was increased in C4, C1, C3 and to a lower extent in C2. This may partly explain the up-regulation of cytokines and interferon pathways in disease clusters. Although plasmacytoid dendritic cells (pDCs) are thought to represent the main IFNα producing cells, no differences were observed between clusters and their reduction was confirmed in peripheral blood of pSS patients when compared to HV[28].

**Cytokine analysis**. We subsequently assessed whether pSS clusters also showed differences in systemic parameters of inflammation, such as cytokines, chemokines and other soluble factors (Fig. 6c and Supplementary Fig. 8). The IFNγ-induced protein (CXCL10/IP-10) as well as CCL8/MCP-2 and TNFα were increased in C1 and C3, i.e. the two main clusters associated with a strong IFN signature. At the same time, IL-1 RII, the decoy receptor for cytokine belonging to the IL-1 family, was down regulated in C1 and C3. Overall, C1 was largely enriched in CXCL13/BLC, IL-6, and IL-1RA. Levels of MMP-8, a protease mainly expressed by neutrophils, were not different from HV in C4 but lower in the other clusters. Of note, many cytokines such as CXCL10/IP-10, CXCL13/BLC, BAFF, and GDF15 were increased in all clusters including C2 when compared to HV. However, no differences between clusters were found for CRP, Fas Ligand, CCL13/MCP-4, CCL4/MIP-1β, CCL17/TARC and TGFβ.

To confirm that patients with an active IFN signature have elevated circulating Type I IFN, we measured levels of IFNα in plasma using Simoa Single Molecule Array Technology in pSS patients and HV. Median levels of IFNα in plasma were 807 (177–1744) fg/ml and 530 (106–1033) fg/ml in C1 and C3, respectively, while circulating levels in the other clusters and HV were close to the lower limit of quantification (Supplementary Fig. 9a). Interestingly, IFNα in serum was positively correlated with the two IFN transcriptomic modules (M1.2 and IFNα module) described in Fig. 3, especially in C1 and to a lesser extent in C3, confirming the Type I IFN signature observed in these patients (Supplementary Fig. 9b). Of note, half of the patients in C2 received antimalarials and previous studies have also shown that hydroxychloroquine use can reduce the levels of circulating Type I[29,30] and Type II[31,32]; IFN z-scores. IFNα in serum was not associated with ESSDAI (Supplementary Fig. 9b) but higher levels of serum IFNα were associated with hematological and biological domains of ESSDAI (Supplementary Data 4).

**Clinical symptoms and serological characteristics**. Patient medical history and disease characteristics including clinical and serological parameters were collected for the 304 pSS patients. Details are displayed in Table 2 and Supplementary Data 5. Patients from C2 had a lower disease duration when compared to patients from other clusters.

Although the Physician Global Assessment (PGA) was collected for the whole population, ESSDAI and ESSPRI were only assessed in expert centers (Barcelona, Brest, Cordoba, Geneva, Hannover, Leuven, Milano, Porto and Szeged) in a subset of 193 and 150 respectively of the 304 pSS studied patients (70/101 and 56/101 from C1, 52/77 and 43/77 from C2, 44/88 and 30/88 from C3 and 27/38 and 21/38 from C4, Supplementary Data 5).

The lowest mean ESSDAI score was observed in C2 and the highest ESSDAI and PGA mean scores in C4 (Table 2, Fig. 7a) but there were no statistically significant differences between the 4 clusters. No clear difference in the ESSDAI components nor in the objective measures of ocular and salivary dryness was observed between the 4 clusters. Moreover, there was no significant difference for the global ESSPRI score and its 3 components (i.e. dryness, pain and fatigue) except between SSA-positive C2 patients who reported lower ESSPRI scores (p-value < 0.001) compared to the SSA-negative patients (Supplementary Data 6).

Statistically significant differences in the distribution of reported arthritis (p-value = 0.016), rate of cancer history (p-value = 0.028), coronary artery disease (p-value = 0.002) and chronic obstructive pulmonary disease (p-value = 0.016) were

observed between the four clusters. (Supplementary Data 7). Interestingly, patients from C4 reported more severe clinical symptoms compared to the 3 other clusters.

Some serological characteristics were significantly different across the 4 clusters, hypergammaglobulinemia (p-value < 0.001) (Table 2), extractable nuclear antigen (ENA) antibodies (p-value <0.001), the presence of serum anti-SSA52/anti-SSA60 autoantibodies (p-value < 0.001) and higher circulating kappa and lambda free light chains (cFLC) (p-value < 0.001) (Fig. 7b, and Supplementary Data 8). C1 and C3 were associated with higher levels of these parameters when compared to C2 and C4. Moreover, C2 and C4 were enriched in patients with glandular manifestations of the disease assessed by a positive focus score in the absence of anti-SSA antibodies (Table 2).

In addition, the levels of rheumatoid factor (p-value < 0.001) and complement C4 fraction levels (p-value = 0.003) were statistically different between the four clusters. C1 was characterized by a higher rheumatoid factor and by a reduced complement C4 fraction levels compared to the other clusters. While some patients presented anti-dsDNA antibodies in C1 and C3 and anti-CCP antibodies in C4, almost none of these autoantibodies were present in the other clusters (Supplementary Data 8).

**Prediction of patient membership to each of the four clusters**. We then developed through machine learning approaches a composite model able to predict, according to a small number of variables, to which of the 4 clusters each patient belongs (see Methods). The proposed composite model was built with a 2-step approach to allocate patient to the right cluster (Supplementary Fig. 10). The final sets of selected features were composed of 10 genes for the C4 prediction model (first step) and 31 genes for the C1, C2, and C3 classification model (second step). The distribution among clusters of the variance stabilizing transformation (vst) normalized expression for all these transcripts is shown in Supplementary Fig. 11. The validation set (Fig. 1 and Table 1) was used for training, due to the heterogeneity of C4 pSS patients in this set, and the composite model was then run on the discovery set. The accuracy of the model was 95.15%, with 99.12% and 95.57%, for the first and the second steps respectively. The confusion matrix, the corresponding discriminant function analysis, and the probabilities to belong to one of the 4 clusters are shown in Fig. 8a, b, and Supplementary Data 9, respectively.

To generalize the composite model, we used an independent inception cohort of 37 pSS patients. After prediction, C1 contained 16 patients (43.2%), C2 6 patients (16.2%), C3 7 patients (18.9%) and C4 8 patients (21.6%). The corresponding discriminant function analysis and the probabilities for a patient to belong to one of the 4 clusters are shown in Fig. 8c and Supplementary Data 10, respectively. We then used the minimal list of 257 discriminative genes signature previously selected in Fig. 1a to generate a heat map with the prediction established by the composite model (Supplementary Fig. 12a). The clusters observed had the same profile than those identified in the discovery set and observed again in the validation set (Fig. 1a), confirming once more the clustering model. Furthermore, the predicted patients showed a distribution of the IFN signatures (Supplementary Fig. 12b) consistent with the one characterizing the identified clusters (Fig. 3). Altogether, these observations strengthen the validation of our composite model.

Finally, in order to allow our model to process other cohorts of patients, we implement an interpolation function based on 6 genes presenting a constant expression across all 4 clusters and HV (Supplementary Fig. 13). The composite model is integrated into an analysis tool available on the laboratory's github repository[33].

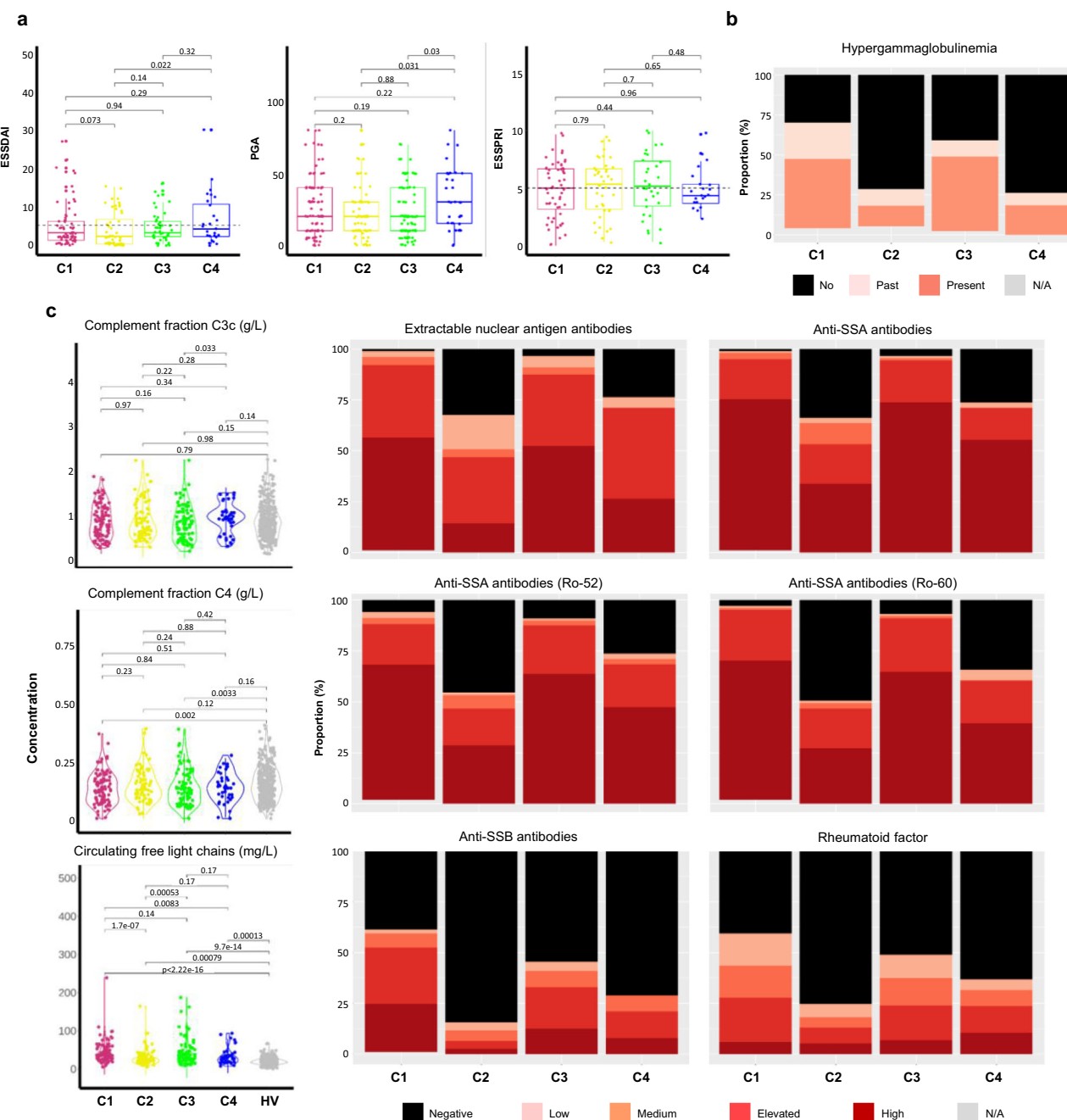

**Fig. 7 Disease activity and serological distributions in the 4 clusters. a** ESSDAI collected for 193 pSS patients (C1: 70, C2: 52, C3: 44, C4: 27), PGA collected for 286 pSS patients (C1: 94, C2: 71, C3: 85, C4: 36,) and ESSPRI collected for 150 pSS patients (C1: 56, C2: 43, C3: 30, C4: 21) distributions are shown in the 4 clusters. Two-tailed pairwise Wilcoxon-rank sum test results are shown. **b** The barplot shows the proportion of past (light orange) or present (orange) hypergammaglobulinemia (C1: 97, C2: 73, C3: 86, C4: 38) in each cluster. **c** Extractable nuclear antigen antibodies, anti-SSA antibodies, anti-SSA antibodies (Ro-52), anti-SSA antibodies (Ro-60), anti-SSB antibodies, rheumatoid factor were performed for 304 pSS patients (C1: 101, C2:77, C3:88, C4:38) and 330 HV and measured in serum, at the same center, using an automated chemiluminescent immunoanalyzer (IDS-iSYS). Barplots show the proportion of concentration level in each cluster (black: negative, light pink: low, orange: medium, red: elevated and dark red: high). Turbidimetry was used for rheumatoid factor (RF), complement fractions C3c and C4 determination and circulating free light chains. Statistical significance is determined by two-tailed pairwise Wilcoxon-rank sum test. Plots show median with error bars indicating ± interquartile range. Patient and HV distribution according to PGA and biological parameters analyzed variable is described in Methods.

## Discussion

Over the last decade, numerous targeted immunomodulatory therapies for pSS have failed to show a benefit in clinical trials, hence no disease-modifying therapy has yet been approved for this disease[34–39]. The heterogeneous nature of pSS and its non-linear development, with flares of activity and subsequent remission associated to a very heterogeneous clinical presentation

may explain clinical trial failures[40]. In this context, there is growing interest in the identification of well-characterized sub-groups of patients, a prerequisite to the identification of mole-cular biomarkers predictive of treatment response[41].

We report herein on a large molecular profiling study carried out in pSS patients, a comprehensive molecular profiling of these patients irrespective of their clinical phenotypes. Previous studies

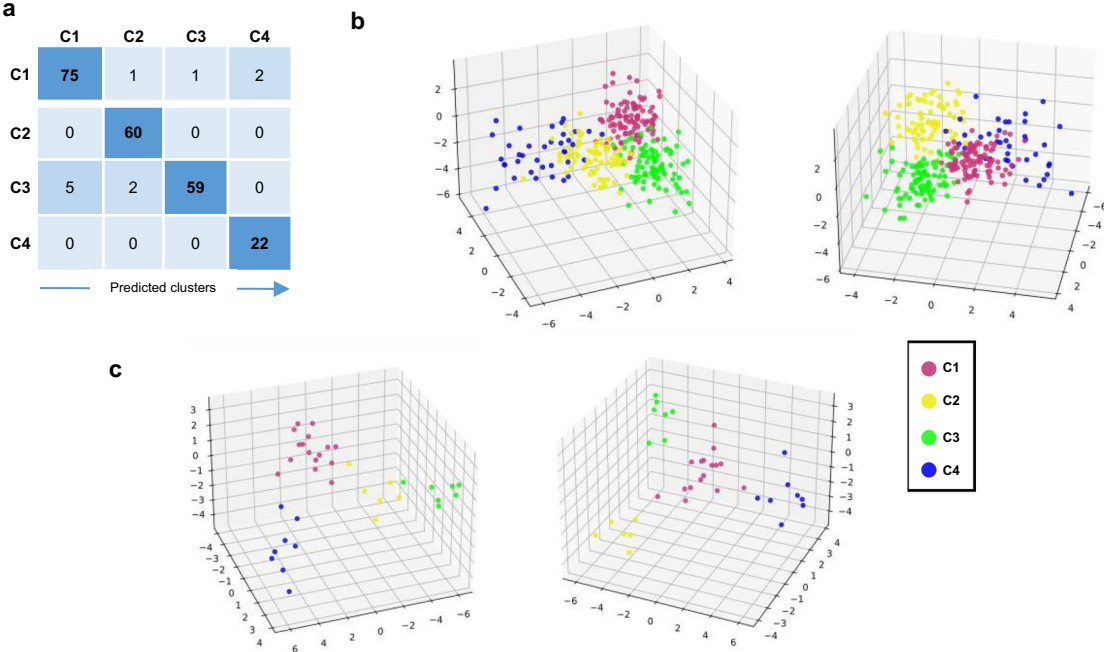

**Fig. 8 Development of a composite model to predict the belonging of a patient to one of the 4 clusters. a** Confusion matrix of the composite model in the discovery cohort performed for 227 pSS patients (C1: 79, C2: 60, C3: 66, and C4: 22) is shown. **b** Discriminant function analysis (DFA) of the predicted patients from the discovery cohort shows clearly separated clusters. Two different views of the same DFA are shown. **c** DFA of the predicted patients from the inception cohort shows clearly separated clusters. Two different views of the same DFA are shown. Thirty-seven pSS patients from the inception cohort were analyzed and predicted as C1: 16, C2: 6, C3: 7, and C4: 8.

in pSS focus particularly on the IFN signaling involvement[11]. Thereby, pSS patients could be stratified in interferon negative, Type I or Type I + II positive subgroups with higher prevalence of anti-SSA and anti-SSB among those with IFN activation without relation with systemic activity. Another group[42] performed a clustering analysis of blood gene expression microarray which classified the 47 pSS patients in three clusters characterized by IFN and inflammation with no discriminant clinical features Moreover, four subgroups of patients with similar patients' clinical characteristics were identified based on absolute cell counts per µL of blood[23]. Lastly, a stratification based on patient clinical phenotypes characterized a posteriori at the molecular level was proposed[43]. These works provide good basis for building a molecular taxonomy of pSS. Our integrative approach using multi-omics and patient clinical characteristics allows going further in understanding pSS heterogeneity.

We identified transcriptional modules allowing to separate pSS patients into four distinct clusters, irrespective of their treatment, reflecting specific patterns of immune dysregulation, with disease activity and patient reported symptom mean scores similar to naturalistic cohorts like ASSESS[44] and UKPSSR [45].

Patients from C2 displayed a healthy-like profile which nonetheless encompasses bona fide pSS patients reporting a similar level of objective symptoms of dryness, pain and fatigue, albeit a lower ESSDAI compared to the 3 other clusters. C2 was also enriched in patients with glandular manifestations of the disease assessed by a positive focus score and no anti-SSA antibodies. A similar cluster was recently described[42] with no increase in the IFN modules and minimal activity of inflammation-related gene modules. Noteworthy, all molecular profiling data reported here were obtained from blood samples which could affect interpretation of some of the results. For example, the reduction of peripheral blood pDCs of pSS patients when compared to HV already reported[28] does not consider that pDC are enriched in the salivary glands and the possibility that tissue sites may be the

major source of IFNα in these individuals[46]. Extending in the future those analyses to the salivary gland will provide a more complete picture of the pathophysiology of the disease, especially in C2.

The three other clusters exhibited significant differences with HV and in particular a prominent IFN gene signature. These findings add to the growing evidence towards a significant role of the IFN pathways in the pathogenesis of systemic and organ-specific disorders including pSS. Whereas Type I IFN were proposed as predominant contributors in the pathogenesis of pSS, a role of Type II IFN in disease pathogenesis has also been highlighted[6,47]. Interestingly, our results show that the IFN signature in the 3 IFN-driven clusters is different. C1 patients had the highest Type I and Type II IFN scores, C3 a higher Type I IFN score than C4, these 2 clusters having similar Type II IFN score. Thus, C4 IFN score was mainly driven by IFN Type II activation. Consequently, C1 and C3 were similar to the IFN cluster recently described by James et al.[42] also associated with high blood protein levels of CXCL10/IP-10.

In line with observed IFN scores, circulating serum levels of IFNα were positively correlated with Type I IFN signature (Supplementary Fig. 9 and Fig. 3) especially in C1 and to a lesser extent in C3. However, levels of IFNα in serum were not correlated with ESSDAI global score, but higher levels of serum IFNα were associated with hematological and biological domains of ESSDAI.

While C1 was mainly driven by IFN, an increase in frequency of B lymphocytes in the blood associated with a significant activation of canonical pathways related to B cell activation such as B cell receptor signaling, and B cell development were observed in C3. Main biological features associated with C3 but also C1 were hypergammaglobulinemia, anti-nuclear antibodies, the presence of serum anti-SSA52/anti-SSA60 autoantibodies and higher cFLC confirming what was already reported in autoantibody-positive pSS patients[21]. Finally, SNPs associated with HLA class II genes

were mainly reported in patients from C1 and C3 presenting a positive IFN signature and high levels of autoantibodies as already shown in SLE[16].

Patients from C4 exhibited a more severe clinical phenotype compared to the others with an inflammatory transcriptomic signature particularly linked to cytokine signaling from the acute phase response. C4 was also characterized by a massive lymphopenia and high levels of neutrophils. The neutrophil-to-lymphocyte ratio (NLR) has been previously shown to correlate with disease activity in systemic autoimmunity[48,49] and elevated NLR are thought to represent a pro-inflammatory state. Indeed, in a study of 483 adult patients with multiple sclerosis, NLR could differentiate between relapsing-remitting and primary progressive multiple sclerosis and predict worsening disability[50]. Further studies are required in pSS to evaluate the importance of this ratio.

Because the main current challenge in clinical trials of new therapies for pSS is the selection of the appropriate patients, we propose here a combination of molecular parameters allowing patient classification by endotypes (Supplementary Fig. 14). We then developed a composite model derived from machine learning, based on the use of a limited number of transcripts from whole blood RNASeq and validated in an independent data set from a pSS inception study, to allow a reanalysis of the previous and ongoing clinical trials to depict predictors of treatment response.

These findings have major implications for the treatment of pSS patients, providing a rationale for both optimal drug positioning and combinations of drugs with complementary mechanisms of action. Specifically, our findings provide a strong rationale for treating patients with either a C1, C3, or C4 profile with inhibitors of type I IFN responses alone or in combination as they support the relevance of B cells as potential therapeutic targets in C3 patients. Trials with B cell depleting antibodies (rituximab) have shown promising results primarily in reducing systemic activity in pSS[51].

Areas requiring further investigation have been identified. First, although our identified cluster gene signatures are strong enough to overcome the disequilibrium in blood cell counts and are not associated with disease duration, except for C2, RNA-Seq analysis is oblivious to sample cell-type composition[52]. Further analyses are on-going, using deconvolution approaches. Second, as hypotheses were derived from a cross-sectional study and a small inception cohort, findings need to be confirmed in longitudinal cohorts to clarify whether patients will stay longitudinally in their initial cluster whatever the disease activity level and the treatments received, or whether treatments decrease disease activity by modifying the extent and scope of gene signaling dysregulations.

Altogether, our results can improve pSS treatment strategies allowing a patient centric approach. This paradigm already implemented in the oncology field will increase the probability of trial successes and boost the development of new efficient drugs against pSS.

## Methods

**Computational tools**. Except when indicated, data analyses were carried out using either an assortment of R system software (http://www.R-project.org, V2.10.1) packages including those of Bioconductor or original R code. R packages are indicated when appropriate. For GWAS analysis, we used Plink, an open-source whole genome association analysis toolset. Machine learning approaches were carried out using python programs (v3.8.5) based on the following modules: scikit-learn, numpy and xgboost.

**Patient population**. The present study was conducted in patients with pSS and HV included in the European multi-center cross-sectional study of the PRECISESADS IMI consortium which involved patients from seven systemic autoimmune

diseases. This study was a pre-planned substudy to be specifically conducted in the pSS population and fulfill the STROBE statements (Supplementary note). Diagnosis of pSS was made according to the 2002 American-European Consensus Group classification criteria, with at least the presence of anti-SSA and/or a positive focus on a minor salivary gland biopsy. Choice of the patient analysis set is detailed in Supplementary Fig. 1a. Recruitment was performed between December 2014 and October 2017 involving 19 institutions in 9 countries (Austria, Belgium, France, Germany, Hungary, Italy, Portugal, Spain and Switzerland). The composite model was validated using transcriptome of 37 pSS newly diagnosed patients recruited in the inception study also obtained from the PRECISESADS consortium. Inception patients were recruited by 10 institutions in Spain, Belgium, France, Italy, Germany and Switzerland. Eligible patients were diagnosed within less than a year since pSS diagnosis.

The two studies (cross-sectional and inception) adhered to the standards set by International Conference on Harmonization and Good Clinical Practice (ICH-GCP), and to the ethical principles that have their origin in the Declaration of Helsinki (2013). Each patient signed an informed consent prior to study inclusion. The Ethical Review Boards of the 19 participating institutions approved the protocol of the cross-sectional study. Moreover, the protocol of the inception study was approved by the ethical committees of the 10 participating institutions. These 10 sites were also participating to the cross-sectional study, therefore these ethical committees reviewed both protocols. The ethical committees involved were: Comitato Etico Milano, Italy; Comité de Protection des Personnes Ouest VI Brest, France; Louvain, Comité d'Ethique Hospitalo-Facultaire, Belgium; Comissao de ética para a Saude—CES do CHP Porto, Portugal; Comité Ética de Investigación Clínica del Hospital Clínic de Barcelona, Spain; Commissie Medische Ethiek UZ KU Leuven/Onderzoek, Belgium; Geschaftsstelle Ethikkommission, Cologne, Germany; Ethikkommission Hannover, Germany; Ethik Kommission. Borschkegasse, Vienna, Austria; Comité de Ética e la Investigación de Centro de Granada, Spain; Commission Cantonale d'éthique de la recherche Hopitaux universitaires de Genève, Switzerland; Csongrad Megyei Kormanyhivatal, Szeged, Hungary; Ethikkommission, Berlin, Germany; Andalusian Public Health System Biobank, Granada, Spain.

The protection of the confidentiality of records that could identify the included subjects is ensured as defined by the EU Directive 2001/20/EC and the applicable national and international requirements relating to data protection in each participating country. The cross-sectional and inception studies are registered in ClinicalTrials.com with respectively number NCT02890121 and number NCT02890134.

For each individual, blood samples as well as biological and clinical information were collected as described in the next Methods sections. For more technical details on sample and data collection, please refer to the main PRECISESADS paper [5].

After quality control on transcriptomics RNAseq data (described below), verification of the ARC/EULAR classification criteria (focus score ≥ 1 foci/mm² and anti-SSA/Ro antibody positivity), and match of the HV to the patients based on age and gender, our final study cohort comprises 304 patients with pSS and 330 HV. This selection is detailed in Supplementary Fig. 1. Among the 304 pSS, 227 (75%) were used for the discovery step and 77 (25%) were kept for validation (Table 1).

**Available data**. High-dimensional omics genotype, transcriptome, DNA methylome and proportions of relevant cell types using flow cytometry custom marker panels were analyzed from whole blood samples. Low dimensional information was obtained from serum samples, including selected serology information such as autoantibodies, cytokines, chemokines and inflammatory mediators. Of note, except for samples collected for flow cytometry analysis, all samples were shipped by the clinical sites to a Central Biobank (Granada) for processing, storage, and onward shipment to the analysis sites, where the various determinations were performed. Flow cytometry was managed at each center on fresh blood after a multi-center harmonization of flow cytometers to ensure mirroring of all instruments[53,54], thereby allowing subsequent integration of all the data obtained across the different sites and instruments. Consequently, all the different omics samples were processed with the same protocols at the same site (RNA-Seq at Bayer, cytokines at UNIMI, autoantibodies and integrated analyses of flow cytometry at UBO, methylome at IDIBELL, GWAS at CSIC which guarantees the high quality of the data generated.

Methods used for RNA sequencing, quality control, data processing, and expression profiling are detailed below and in Supplementary Fig. 1c.

**RNA-Seq**. Methods used for RNA sequencing, quality control, data processing, and expression profiling are detailed below and in Supplementary Fig. 1c. Total RNA was extracted from whole blood samples collected in Tempus tubes using Tempus Spin technology (Applied Biosystems). 1857 samples were processed in batches of 384, randomized to four 96-well plates with respect to patient diagnosis, recruitment center and RNA extraction date. The samples were depleted in alpha- and beta-globin mRNAs using globinCLEAR protocol (Ambion) and 1 μg of total RNA was used as input. Subsequently, 400 ng of globin-depleted total RNA was used for library synthesis with TruSeq Stranded mRNA HT kit (Illumina). The libraries were quantified using qPCR with PerfeCTa NGS kit (Quanta Biosciences), and equimolar amounts of samples from the same 96-well plate were pooled. Four

pools were clustered on a high output flow cell (two lanes per pool) using HiSeq SR Cluster kit v4 and the cBot instrument (Illumina). Subsequently, 50 cycles of single-read sequencing were performed on a HiSeq2500 instrument using and HiSeq SBS kit v4 (Illumina). The clustering and sequencing steps were repeated for a total of three runs in order to generate sufficient number of reads per sample. The raw sequencing data for each run were preprocessed using bcl2fastq software and the quality was assessed using FastQC tools. Cutadapt[55] was used to remove 3′ end nucleotides below 20 Phred quality score and extraneous adapters, additionally reads below 25 nucleotides after trimming were discarded. Reads were then processed and aligned to the UCSC Homo sapiens reference genome (Build hg19) using STAR v2.5.2b[56]. 2-pass mapping with default alignment parameters were used. To produce the quantification data, we used RSEM v1.2.31[57] resulting in gene level expression estimates (Transcripts Per Million, TPM and read counts).

For sample filtering, samples were filtered in at least one of the following situations: (i) the total sum of count is too low (<5000,000), (ii) they were extracted with another method than Tempus Spin, and (iii) the RIN (RNA Integrated Number) value of the sample is below 6.5, (iv) samples with RNAseq inferred gender inconsistent with clinical data, and (v) there was a disagreement between genotypes inferred from RNA-Seq and those obtained from GWAS genotyping.

For normalizations and batch correction, read counts were normalized by the variance stabilizing transformation vst function from DESeq2 (v1.30.0) R package[58]. To reduce the effect of the RIN, a correction was applied using the ComBat function from sva (v3.38.0) R package[59], after categorization of RIN values into 7 classes: (7.5, 8], (8.5, 9], (9.5, 10], (8, 8.5], (7, 7.5], (9, 9.5], (6.5, 7].

For Gene filtering, among the 55,771 genes detected in the data, those with 0 count over all the samples or having an expression level below 1 in more than 95% were filtered. At the end, our final RNA-Seq data comprises 16,876 genes. This selection is detailed in Supplementary Fig. 1.

**Molecular subgroups discovery.** Our rational was to produce a robust classification scheme and to ensure the greatest possible homogeneity within identified subgroups. To this aim, subgroup discovery was based on the pre-processed RNA-seq data of the discovery set (after vst transformation). We implemented a strategy already applied in breast cancer that iterates unsupervised and supervised steps, which was, therefore, designated as "semi-supervised" approach[8]. It is described hereafter and summarized in Supplementary Fig. 2.

Step 1: Unsupervised gene selection

The coefficient of variation ($CV_g = \frac{\sigma_g}{\mu_g}$, with $\sigma_g$ is the standard deviation of the gene $g$ and $\mu_g$ the mean of the gene $g$ estimated on discovery population) and its robust version ($rCV_g = \frac{\gamma_g}{\mu_g}$, with $\gamma_g$ is the median absolute deviation) were calculated for each gene. Both were highly concordant. The top 25% most variants were selected to perform the subsequent clustering analysis.

Step 2: Robust consensus clustering

To determine the number of clusters, a consensus clustering between three methods was performed: (i) Agglomerative Hierarchical Clustering (hclust function from stats v4.0.2 R package) with Pearson correlation as a similarity measure and the Ward's linkage method, (ii) K-means clustering (kmeans function from stats R package) with 4 groups and (iii) Gaussian mixture clustering (mclust function from mclust v5.4.6 R package).

Step 3: Identification of molecular signature

A supervised analysis was performed on the 149 patients with consistent cluster assignments between the three clustering methods (considered as "core" molecular profiles), in order to identify the most discriminating signature of the 4 clusters. The first signature of set of 3577 genes was selected from a classical one-way ANOVA (FDR < 1e-10), and then reduced by Random Forest to 257 top discriminating genes (randomForest function from randomForest v4.6-14 R package[60]).

Step 4: Robustness classification

To validate the robustness of our clustering, we re-applied Step 2 on our discovery set and on the final signature.

Step 5: Classification of discordant patients

Patients assigned to different groups with the 3 clustering methods were assigned to one of the 4 clusters by applying a distance-to-centroid method based on Pearson correlation.

**Molecular subgroup validation.** Validation datasets were independently classified in the pSS molecular subgroups by applying a classical distance-to-centroid approach based on correlation. Following the same approach, HV did not constitute a separate cluster but mainly matched with C2 (0.5% in C1, 93% in C2, 4% in C3, and 2.5% in C4) pSS molecular subgroups by applying a classical distance-to-centroid approach based on correlation. The final clustering (without HV) is represented with heatmap using the Heatmap function from ComplexHeatmap (v2.6.2) R package. Clusters are separately constrained for better visualization. This method allows to spotlight heterogeneous intra-clusters. The principal component analysis (PCA) representation will explore the clearly defined clusters and the matching between C2 and HV.

Half of the pSS patients was treated with either anti-malarial, immunosuppressant, or steroids at the time of the visit. When compared to the 3 other clusters, we observed higher proportion of treated patients in C4. To investigate the impact of the treatment on the clustering, we compared treated patients and untreated patients. For this, we apply a hierarchical clustering on treated patients and untreated patients and compare the cluster distribution. The heatmap (Supplementary Fig. 4) of treated vs untreated patients were highly similar which shows that the final clustering is not driven by treatments.

**Enrichment analysis.** Enrichment analysis was performed by applying a two-tailed Fisher-exact test[61] against different sources of gene modules or pathways: (i) 3 strongly upregulated IFN-annotated modules from[10] (M1.2, M3.4, and M5.12) determined from peripheral blood transcriptomic data with for each a distinct activation threshold, (ii) genes preferentially induced by IFNα or IFNγ identified by[10], (iii) canonical pathway from Ingenuity Pathway Analysis (IPA, Release Date: 2020-06-01), (iv) repertoire recently established on an expended range of disease and pathological states (382 transcriptome modules based on genes co-expression patterns across 16 diseases and 985 unique transcriptome profiles) by [9].

**Differential gene expression analysis.** To identify genes differentially expressed between pSS subgroups and HV, we performed a linear model (lmFit function from limma v3.46.0 R package[62]) on vst transformation gene expression dataset. Resulting p-values were adjusted for multiple hypothesis testing and filtered to retain DE genes with FDR adjusted p-value ≤ 0.05 and a |Fold-Change (FC)| ≥ 1.5.

**Genome-wide association study.** Genome-wide association studies (GWAS) were performed for each pSS subgroups (C1: 101, C2: 77, C3: 88, and C4: 38) versus 330 HV. After DNA extraction, the samples were genotyped using HumanCore v1.0 and Infinium CoreExome-24 v1.2 genome-wide SNP genotyping platform (Illumina Inc., San Diego, CA, USA). Individuals were excluded on the basis of incorrect gender assignment, high missingness (>10%), non- European ancestry (<55% using Frappe15 and REAP), and high relatedness (PLINK v1.9[45], pi_hat >0.5)[63]. Genotypes were filtered before imputation due to high missingness (>2%), Hardy–Weinberg equilibrium (HWE) < 0.001, minor allele frequency (MAF) <1%, and AT/CG changes with MAF >40%. PLINK v1.9[45] was used to carry out quality control (QC) measures, genotype data filtering. The basic association for a cluster trait locus, based on comparing allele frequency between patients from each cluster vs HV, was also obtained with this toolset thanks to computational resources from the Roscoff Bioinformatics platform ABiMS. Genotypes were phased using Eagle v2.3 and imputed using Minimac3 against the HRC v1.1 Genomes reference panel from the Michigan Imputation Server platform. Genotypes were filtered after imputation to have HWE p-value > 0.001, MAF > 1% and imputation info score > 0.7 and resulted in 6,664,685 imputed genotypes. Statistical analysis of association for each cluster versus HV was performed by logistic regression under the additive allelic model. The GWAS significant level was fixed at p-value < $5 \times 10^{-8}$. SNP annotations and Manhattan plot were obtained using the web-based tool SNP snap from the Broad Institute[64] and qqman (v0.1.8)[65] R packages respectively.

**Methylation.** Whole blood methylation analysis was performed for 226 pSS patients (C1: 81, C2: 57, C3: 62, and C4: 26) and 175 healthy volunteers (HV). DNA was extracted using a magnetic-bead nucleic acid isolation protocol (Chemagic DNA Blood Kit special, CHEMAGEN) automated with chemagic Magnetic Separation Module I (PerkinElmer) from K2EDTA blood tube (lavender cap, BD Vacutainer) of 10 ml (extractions were performed on 3 ml). 2 µg of DNA were sent for DNA methylation assay. The samples were analyzed using Infinium Human Methylation 450 K BeadChip (Illumina, Inc., San Diego, CA, USA) which covers more than 400,000 CpG sites. DNA samples were bisulfite-converted using the EZ DNA methylation kit (Zymo Research, Orange, CA, USA). After bisulfite conversion, the remaining assay steps were performed following the specifications recommended by the manufacturer. The array was hybridized using a temperature gradient program, and arrays were imaged using a BeadArray Reader (Illumina Inc., San Diego, CA, USA). Sample QC and functional normalization were completed using minfi (v3.3) R package[66]. Briefly, during QC steps, subjects were removed based on outliers for methylated vs unmethylated signals, deviation from mean values at control probes, and high proportion of undetected probes (using minfi default parameters). DNA methylation probes that overlapped with SNPs (dbSNPs v147), located in sexual chromosomes or considered cross-reactive were removed. Additionally, only probes quality controlled and shared between both arrays were used in the subsequent analysis (368,607 probes). Measure of methylation level (B values) were produced for each CpG probe and ranged from 0 (0% molecules methylated at a particular sites) to 1 (100% molecules methylated).

To identify differentially methylated positions (DMPs) between HV and each pSS subgroups (C1 to C4), the champ.DMP function of ChAMP (v2.18.3) R package[67] was implemented doing pairwise comparison between each cluster and HV. Many Δ-beta thresholds were described in the literature and the most frequently used for whole blood studies in autoimmune diseases were 0.05 (5% difference) and 0.1 (10% difference). In order to fix the best threshold for our study, we tested the values of 0.05, 0.075, 0.1, and 0.15 for the absolute ΔBeta. Supplementary Data 11 presents the numbers of DMPs and genes obtained with these different thresholds.

Then, we decided to analyze the data in two steps: the first step with a significant adjusted p-value (Benjamini Hochberg) at 0.1 and an absolute ΔBeta > 0.075. We assumed that a threshold of 0.05 was too low and it would have been

very difficult to interpret the signification of these defects in methylation for C4. If we had applied a ΔBeta threshold of 0.1 in the first intention, we could have missed DMPs. In the second step in order to identify the most robust and significant signature of hypo and hyper methylated genes, a significant adjusted *p*-value (Benjamini Hochberg) at 0.1 and an absolute ΔBeta > 0.15 were applied.

For network viewing, we tested gene lists onto the STRING 9.1 Network of Known and Predicted Protein–Protein Interactions (http://string-db.org/)[68].

**Flow cytometry.** Multi-parameter flow cytometry analyses have been performed in eleven different centers from the PRECISESADS consortium. Therefore, the integration of all data in common bioinformatical and biostatistical investigations has required a fine mirroring of all instruments[54]. The calibration procedure elaborated to achieve this prerequisite and the antibody panels used have been previously described[53].

The antibody panels, specificities, and clones used are shown in Supplementary Fig. 15a.

The strategy developed to avoid any redundancy in the different cell subsets and to increase the accuracy of the phenotypes has been automated by AltraBio (Lyon, France). The generated automatons have been validated in a preliminary study on 300 patients comparing data from automated gating to data manually gated by the same operator (coefficient of correlation 0.9996). The gating strategy was as follows: after exclusion of debris, dead cells and doublets, frequencies and absolute numbers of CD15hiCD16hi neutrophils, CD15hiCD16+ eosinophils, CD14+CD15hi LDGs, CD14++CD16− classical monocytes, CD14++CD16+ intermediate monocytes, CD14+CD16++ non classical monocytes, CD3+ T cells (with CD4+CD8−, CD4+CD8+, CD4−CD8+, CD4−CD8− T cell subsets), CD19+B cells, CD3−CD56+ NK cells (with CD16loCD56hi and CD16hiCD56lo NK cell subsets), CD3+CD56+ NK-like cells, Lin-HLA-DR+ DCs (with CD11c−CD123+ pDCs, CD11c+CD123− mDCs (with CD141−CD1c+ mDC1, CD141+CD11c− mDC2 and CD141−CD1c− mDC subsets)) and CD123+HLA-DR− basophils were automatically extracted from FCS and LMD files of 283 patients and 309 HV and sent in an Excel flow cytometry workflow. The mean distribution of blood cell subsets in frequency (0–100%) and absolute numbers by clusters are compared using a Kruskal–Wallis test.

Gating strategies of the automatons are shown in Supplementary Fig. 15b. For all instruments, the data from the flow cytometry files are analyzed with a similar strategy by one automaton for panel 1 and another automaton for panel 2, and then specifically for each instrument from the gate [S4] to account for the variability of FSC and SSC signals. The desired cell populations are identified by gating strategies identical for all instruments for panel 1 and panel 2 stainings. The mean distribution of blood cell subsets in frequency and absolute numbers are shown in Supplementary Data 12 and 13, respectively.

**Cytokines.** Cytokines were measured on serum samples. CXCL13/BLC, FAS Ligand, GDF15, CXCL10/IP-10, CCL8/MCP-2, CCL13/MCP-4, CCL4/MIP-1β, MMP-8, CCL17/TARC, IL-1 RII, TNF RI, and IL1-Ra were measured using the Luminex system. The 12-analyte customized panel was built using human pre-mixed multi-analyte Luminex assay (R&D Systems). Samples were thawed on the day of analysis and tested in batches. Soluble MMP-2, CRP, TNFα, IL-6, BAFF, and TGFβ were measured using ELISA assay. Descriptive statistics are shown in Supplementary Data 14. We measured levels of IFNα in plasma using Simoa Single Molecule Array Technology. Results were calculated referring to a standard curve created using a four parameters logistic curve fit and were expressed as pg/ml. For more technical details on sample and data collection, please refer to the main PRECISESADS study[5]. The differential cytokine concentration between subgroups vs HV was performed using a one-way ANOVA followed by post-hoc Tukey's test (function ghlt from multicomp multcomp v1.4-13 R package[69]). The z-score indicate the direction of the concentration between the cluster and the HV. A z-score > 0 means that the cluster has an overexpression compare to HV. A z-score < 0 means that the cluster has a lower expression compare to HV (Fig. 6). Concentration distribution by subgroup is represented in Supplementary Fig. 8. Two-tailed pairwise Wilcoxon-rank sum tests have been computed.

**Autoantibodies.** Autoantibodies (Extractable nuclear antigen antibodies, anti-SSA antibodies, anti-SSA antibodies (Ro-52), anti-SSA antibodies (Ro-60), Anti-SSB antibodies), were measured in serum using an automated chemiluminescent immunoanalyzer (IDS-iSYS). After processing, the final result is indicative of the concentration of the specific autoantibody present in the sample. Rheumatoid factor (RF), complement C3c, C4, and individualized (kappa, lambda) free light chains (Combilite and freelight, respectively) were measured in serum using a turbidimetric immunoassay method according to manufacturer's recommendations (SPAPLUS analyser). For more technical details on sample and data collection, please refer to the main PRECISESADS study[5]. Autoantibodies and RF distribution have been described by concentration level (Negative/Low/Medium/Elevated/High) and a Fisher's exact test was applied to compare the proportion and the concentration across the 4 clusters. Complements C3 and C4 and circulating free light chains have been described in continued concentration expressed in g/L and mg/L respectively and a Kruskal–Wallis test was applied to compare the concentration level across the 4 clusters. Descriptive statistics are described in Supplementary Data 8.

**Clinical data.** Clinical data on 304 patients with pSS and 330 HV describing the disease phenotype was collected using an electronic case report form (eCRF). A working group of experts on systemic autoimmune diseases was established and the desired items were selected via a Delphi technique. A final set of items was created, digitalized and pilot tested divided into 8 domains (constitutional symptoms, gastrointestinal, vascular, heart and lung, nervous system, skin and glands, musculoskeletal, therapy). After the confirmation of patient inclusion, clinical data were collected including patient's age, sex, ethnicity, dates of first disease manifestation (disease onset), clinical and biological characteristics at baseline, the physician global assessment of disease activity, comorbidity, and current use of treatments.

Another working group of pSS pathology experts was established to select pSS disease-specific items, mainly pSS disease activity scales like ESSDAI and its components, and ESSPRI and its components. These items were collected on a pSS sub-population ($n = 193$).

To characterize pSS subgroups, association test was performed with clinical data. A two-tailed Fisher's exact test (fisher.test function from stats R package) or chi-square test (chisq.test function from stats R package) as appropriate was applied to evaluate the association between the pSS supbgroups and a qualitative clinical factor. A Kruskal–Wallis test (Kruskal.Wallis function from stats R package) was used to evaluate the association between pSS subgroups and quantitative clinical variables.

**Development of the composite model for cluster prediction.** This feature selection process is composed of two distinct parts: (i) identify a subset of genes potentially interesting to predict the 4 clusters, (ii) use these previously identified subsets to actually craft a prediction model and extract the features used by the model to increase its precision. In the first part, with FC ≥1.5 and FDR ≤0.05, we selected the DEGs according to the following 7 combinations: C2 vs C1, C3 vs C1, C4 vs C1, C4 vs C2, C3 vs C2, C4 vs C2, C4 vs C3. We identified 14,240 and selected those common to all combinations representing 1154 DEGs.

We used the Boruta algorithm[70] on all dataset (discovery and validation sets) to extract features that significantly contributed to predict the patient's cluster.

The algorithm started to extend the dataset by adding copies of each feature in the original dataset. These features were called "shadow features" and consisted in random permutation of the modality of the original feature, in order to remove any correlation with the target variable, in our case, the cluster assignment. Once shadow features were crafted, a random forest classifier was run on the whole dataset and z-scores were computed for all features (real and "shadow"). Shadow features were then sorted according to their z-score and the maximum score was kept in memory as a threshold. The algorithm assigned a hit to each real feature that had a z-score above this threshold. Finally, Boruta marked the features which had a z-score significantly lower than the shadow with maximum z-score as "unimportant" and removed them from the dataset, before removing all shadow features and returning a clean dataset.

This process allowed us to identify variables in the dataset that were significantly more contributing to the classification problems than noisy variables and random artefacts emulated by the original variable modality permutation, ensuring the use of robust features for the second step of our feature selection strategy.

The relatively small size and heterogeneity of C4 in comparison to the other clusters can impact the feature selection process, therefore we chose to solve two classification problems: (i) identify C4 versus all clusters, (ii) discriminate between C1, C2, and C3.

The operation was performed twice: one to predict C4 cluster versus all other clusters and one to discriminate between C1, C2, and C3. In both cases, the algorithm ran over 100 iterations with a max depth of 5 and balanced classes for initializations of random forests.

The two sets of selected features were respectively composed of 255 genes for the C4 prediction dataset and 597 genes for the C1, C2, and C3 prediction dataset.

We then used xgboost-tree[71] approach, to train a model on the dataset with a binary logistic objective function to predict C4 vs all (using the 255 genes previously identified by the Boruta algorithm) and to extract features that have been used by the algorithm to craft the decision tree of the model.

The model can be summarized by $\hat{y}_i = \sum_{k=1}^{K} f_k(x_i), f_k \in F$ where $\hat{y}_i$ is the cluster prediction for the patient $i$, $x_i$ the vector describing the patient $i$ (composed of the selected features), $F$ the set of estimators for the model (4 in our case, one for each cluster) and $K$ the number of trees by estimator which is 3 for C4 and 4 for C1, C2, and C3. In this context, $f_k$ refers to the tree number $k$ of the estimator $f$ where $f \in F$. $K$ has been manually refined in order to find a compromise between good predictive performance and a low complexity model.

We performed the same approach with a softmax objective function in a multi-classification context to predict the C1, C2, and C3 cluster based on the 597 features previously highlighted by Boruta for this specific classification problem.

The final sets of selected features were composed of 10 genes for the C4 prediction model and 31 genes for the multi-classification (C1, C2, or C3) model (Supplementary Fig. 10). The accuracies of the models, during the training phase perform on the validation set (Table 1) were 94.81% for the C4 prediction model and 96.72% for the multi-classification model.

We then created a composite model, using the combinatorial results of the C4 predictor model and the multi-classification model to predict all 4 clusters on the patients of the discovery set.

Patients were first evaluated by the C4 predictor model. If C4 was not assigned, the patients were evaluated by the multi-classification model.

In order to allow our model to process other cohorts of patients we implemented an interpolation function described by (2). We selected 6 genes with FC ≤ 1.1 and FDR ≥ 0.05 based on their constant expression across all 4 clusters and HV. Their expressions were between 4 and 14 vst normalized counts [*SPIRE* (4), *NUP210L* (6), *GATAD1* (8), *HVCN1* (10), *ENO* (12), and *FLNA* (14)] (Supplementary Fig. 13). This set of genes was denoted G. The interpolated value of a gene x, $I(x)$ was computed as $I(x) = I(a) + (I(b) - I(a)) \times \frac{x-a}{b-a}$ with a and b representing the vst normalized expression value of two genes such as genes $a, b \in G$, $a < x < b$ and $b \neq a$.

The composite model is integrated into an analysis tool available[33] and the pseudocode description is reported in Supplementary Fig. 16.

**Reporting summary**. Further information on research design is available in the Nature Research Reporting Summary linked to this article.

## Data availability

All data included in our study is available upon request at ELIXIR Luxemburg, except the GWAS data that cannot be anonymized, with the permanent link: https://doi.org/10.17881/th9v-xt85 and access procedure is described on the ELIXIR data landing page. The PRECISESADS Consortium committed to secure patient data access through the ELIXIR platform. This commitment was formerly given by written to all patients at the end of the project and to the involved Ethical Committees. The future use of the Project database was framed according to the scope of the patient information and consent forms, where the use of patient data is limited to scientific research in autoimmune diseases. ELIXIR reviews applicants requests and prepares Data Access Committee's decisions on access to Data, communicates such decisions to the Data Providers, who have 10 days to exercise their right to veto; otherwise access is granted to the User.

## Code availability

Except when indicated, data analyses were carried out using either an assortment of R system software (http://www.R-project.org, V4.0.1) packages including those of Bioconductor or original R code. R packages are indicated when appropriate. For GWAS analysis, we used Plink, an open-source whole genome association analysis toolset. Machine learning approaches were carried out using python programs (v3.8.5) based on the following modules: scikit-learn, numpy, and xgboost. The composite model designed to predict the patient's cluster is integrated into an analysis tool available on the laboratory's github repository at the following address: https://lbai-infolab.github.io/SjTree/(33).

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

## Acknowledgements

The research leading to these results has received support from the Innovative Medicines Initiative Joint Undertaking under the Grant Agreement Number 115565 (PRE-CISESADS project), resources of which are composed of financial contribution from the European Union's Seventh Framework Program (FP7/2007–2013) and EFPIA companies' in-kind contribution. LBAI was supported by the Agence Nationale de la Recherche under the "Investissement d'Avenir" program with the Reference ANR-11-LABX-0016-001 (Labex IGO) and the Région Bretagne. The authors would like to particularly express their gratitude to the patients, nurses, technicians and many others who helped directly or indirectly in the consecution of this study. They are grateful to the Institut Français de Bioinformatique (ANR-11-INBS-0013), the Roscoff Bioinformatics platform ABiMS (http://abims.sb-roscoff.fr) for providing computing and storage resources and the Hypérion platform at LBAI (Brest, France) for flow cytometry facilities. Finally, this work is now supported by ELIXIR Luxembourg via its data hosting service.

## Author contributions

P.S., C.L.D., E.D., B.C., S.H., and C.B. performed the computational studies and carried out the analysis, N.F. performed the computational studies and developed the composite model, C.J., G.B., G.D., PRECISESADS Flow Cytometry Consortium, E.B., J.M., A.B., Z. M. R.L., M.O.B. performed the experimental studies. V.D.P., D.C., A.S., S.J.J., N.B.P., I.R. P., E.D.L., L.B., C.C., L.K., T.W., and PRECISESADS Clinical Consortium contributed to the recruitment of patients. S.C.G., L.X., M.G., P.M. contributed to the edition of the manuscript, MEAR supervised the PRECISESADS consortium, L.L. and J.O.P. supervised the work and wrote the manuscript. All the authors have approved the content of this paper and its related supplementary files and have agreed to the Nature Communications submission policies.

## Competing interests

While engaged in the research project, R.L., F.M., and Z.M. were regular employees of Bayer A.G. At present, R.L. and Z.M. are regular employees of Nuvisan ICB GmbH, a company providing contract research services. P.S., S.H., S.C.G., L.X., M.G., P.M., and L. L. were regular employees of Institut de Recherches Internationales Servier at the time of the research project. B.C., C.B., and E.D. were phD students financed by Institut de Recherches Internationales Servier when they contributed to the research project. All other authors confirmed signing the ICMJE form for Disclosure of Potential Conflicts of Interest and none of them have any conflict of interest related to this work.

## Additional information

## PRECISESADS Clinical Consortium

Lorenzo Beretta[8], Barbara Vigone[8], Jacques-Olivier Pers[2,3], Alain Saraux[2,3], Valérie Devauchelle-Pensec[2,3], Divi Cornec[2,3], Sandrine Jousse-Joulin[2,3], Bernard Lauwerys[15], Julie Ducreux[15], Anne-Lise Maudoux[15], Carlos Vasconcelos[16], Ana Tavares[16], Esmeralda Neves[16], Raquel Faria[16], Mariana Brandão[16], Ana Campar[16], António Marinho[16], Fátima Farinha[16], Isabel Almeida[16], Miguel Angel Gonzalez-Gay Mantecón[17], Ricardo Blanco Alonso[17], Alfonso Corrales Martínez[17], Ricard Cervera[6], Ignasi Rodríguez-Pintó[6], Gerard Espinosa[6], Rik Lories[7], Ellen De Langhe[7], Nicolas Hunzelmann[18], Doreen Belz[18], Torsten Witte[11], Niklas Baerlecken[11], Georg Stummvoll[19], Michael Zauner[19], Michaela Lehner[19], Eduardo Collantes[5], Rafaela Ortega-Castro[5], Ma Angeles Aguirre-Zamorano[5], Alejandro Escudero-Contreras[5], Ma Carmen Castro-Villegas[5], Yolanda Jiménez Gómez[5], Norberto Ortego[20], María Concepción Fernández Roldán[20], Enrique Raya[21], Inmaculada Jiménez Moleón[21], Enrique de Ramon[22], Isabel Díaz Quintero[22], Pier Luigi Meroni[13], Maria Gerosa[13], Tommaso Schioppo[13], Carolina Artusi[13], Carlo Chizzolini[9], Aleksandra Zuber[9], Donatienne Wynar[9], Laszló Kovács[10], Attila Balog[10], Magdolna Deák[10], Márta Bocskai[10], Sonja Dulic[10], Gabriella Kádár[10], Falk Hiepe[23], Velia Gerl[23], Silvia Thiel[23], Manuel Rodriguez Maresca[24], Antonio López-Berrio[24], Rocío Aguilar-Quesada[24], Héctor Navarro-Linares[24], Yiannis Ioannou[25], Chris Chamberlain[26], Jacqueline Marovac[26], Marta Alarcón Riquelme[4] & Tania Gomes Anjos[4]

[15]Pôle de pathologies rhumatismales systémiques et inflammatoires, Institut de Recherche Expérimentale et Clinique, Université catholique de Louvain, Brussels, Belgium. [16]Centro Hospitalar do Porto, Port, Portugal. [17]Servicio Cantabro de Salud, Hospital Universitario Marqués de Valdecilla, Santander, Spain. [18]Klinikum der Universitaet zu Koeln, Cologne, Germany. [19]Medical University Vienna, Vienna, Austria. [20]Complejo hospitalario Universitario de Granada (Hospital Universitario San Cecilio), Granada, Spain. [21]Complejo hospitalario Universitario de Granada (Hospital Virgen de las Nieves), Granada, Spain. [22]Hospital Regional Universitario de Málaga, Málaga, Spain. [23]Charite, Berlin, Germany. [24]Andalusian Public Health System Biobank, Granada, Spain. [25]UCB Pharma (PRECISESADS Project office), Slough, UK. [26]Chromatin and Disease Group, Bellvitge Biomedical Research Institute (IDIBELL), Barcelona, Spain.

## PRECISESADS Flow Cytometry Consortium

Christophe Jamin[2,3], Concepción Marañón[4], Lucas Le Lann[2], Quentin Simon[2], Bénédicte Rouvière[2,3], Nieves Varela[4], Brian Muchmore[4], Aleksandra Dufour[9], Montserrat Alvarez[9], Carlo Chizzolini[9], Jonathan Cremer[7], Ellen De Langhe[7], Nuria Barbarroja[5], Chary Lopez-Pedrera[5], Velia Gerl[23], Laleh Khodadadi[23], Qingyu Cheng[23], Anne Buttgereit[12], Zuzanna Makowska[12], Aurélie De Groof[14], Julie Ducreux[14], Elena Trombetta[8], Tianlu Li[26], Damiana Alvarez-Errico[26], Torsten Witte[11], Katja Kniesch[11], Nancy Azevedo[15], Esmeralda Neves[15], Sambasiva Rao[27], Pierre-Emmanuel Jouve[28] & Jacques-Olivier Pers[2,3]

[27]Sanofi Genzyme, Framingham, MA, USA. [28]AltraBio SAS, Lyon, France.

