## [Peer Review File · Nature Communications]

Reviewers' Comments:

Reviewer #1:

Remarks to the Author:

A major barrier to development of effective therapies for Sjogren's disease is lack of understanding of disease heterogeneity at the molecular level. Soret and colleagues use peripheral blood RNA sequencing data from 304 Sjogren's patients to define four molecular subgroups of patients and subsequently annotate the groups for associated SNPs using GWAS, global methylome, comprehensive flow cytometry, a limited number of serum cytokines, as well as comprehensive clinical data. The project leverages harmonized data from the PRECISESADS Clinical Consortium, with contributions from multiple institutions in multiple European countries. Enrichment analyses using published gene modules and Ingenuity Pathways Analysis further describe the clusters. The authors then use the most variable transcripts to develop a composite model able to predict cluster membership. The authors finally present an accessible interpolation function to enable other investigators to use this prediction tool with their own datasets.

This is a significant study that may advance the objective of developing effective therapies for Sjogren's disease and can immediately be applied to existing clinical trial data if appropriate PBMC RNA-Seq data is available in order to stratify responses by subgroup membership. The data and harmonization across centers, quality control and analyses in general appear to be robust. Extensive supplementary data and methods support the findings.

There are concerns and limitations of the study that should be more carefully considered (Points 1, 2, 4). Some of the results are over-simplified (Point 5). There are several more minor edits that could be made to enhance clarity.

1. The composite predictive model was developed/trained using the smaller inception cohort then tested on the larger cross-sectional cohort. Yet, the approach used to select which genes were to be used in the composite predictive model utilized RNA-Seq data from all of the subjects. Whether this constitutes a true test of the model is thus unclear. The study would be further strengthened by testing the composite predictive model using data from different subjects that were not used to develop the model. In their favor, the authors have facilitated the capacity of others to test the model in their own datasets.
2. Cluster 1 is enriched in disease-associated SNPs (n=35). To what extent if any do eQTLs explain the gene expression responsible for delineating this cluster?
3. Figure 2 is unclear. In particular, the color scheme in Figure 2B is too complex (too many colors). The authors may consider using letters of the alphabet in lieu of colors. The white boxes in Figure 2A are not adequately described. X axes (1-42) in Figure 2A are not adequately described. Annotation of the most important features of Figure 2A with the abbreviations in Figure 2B would be helpful. The Supplementary Figure 5, on the other hand, is clear and easy to interpret.
4. A limitation of the study includes the use of data only from peripheral blood. This is mentioned in the Discussion. This limitation affects interpretation of some of the results. For example, on lines 327-329, the authors seem to suggest that a reduction of peripheral blood pDC is incongruent with the notion that pDC are thought to be the major interferon alpha producers. The authors do not consider published data showing that pDC are enriched in the salivary glands of Sjogren's patients and the possibility that tissue sites may be the major source of interferon alpha in these individuals.
5. There is considerable overlap of interferon signatures among Clusters 1, 2 and 4. The description of Cluster 4 as "rather of Type II" interferon cluster (Abstract line 61) is an oversimplification, as this cluster also contains strong enrichment of Type 1 interferon modules. Similarly, characterization of Cluster 3 as the B lymphocyte patient cluster may be over-simplified. Other clusters show some enrichment in B cell features, including IPA B cell signaling pathways (also enriched in Clusters 1 and 4, Suppl Table 2), hypergammaglobulinemia (high in Clusters 1 and 3), elevated RF (also observed in Cluster 1), circulating free light chains (elevated in all clusters vs. HV), SSA/SSB positivity (prevalent in Clusters 1 and 3), ESSDAI biological domain (elevated in Clusters 1 and 3). Over-simplifying description of the patient clusters could unintentionally create confusion in the field.
6. Provision of complete lists of differentially expressed genes would be desirable to enhance accessibility of the study results.

7. Mention of 2016 ACR/EULAR criteria in the opening sentence of the Results section is confusing, as the Methods state that AECG criteria were used for this study.
8. A clear definition of disease duration used in Table 1 is needed. Is this from date of diagnosis or date of first symptom(s), as both fields of data are listed in the Supplementary Methods?
9. Inclusion of Healthy Volunteer data in Figure 6C would be helpful.
10. Figure 5B colors denoting 5'UTR and IGR and difficult to distinguish.
11. In Suppl Figure 6, the significance designations in the figures are too small to be legible. These should be enlarged or the data presented in tables.
12. In Suppl Figure 13, whether the flow cytometry items utilize frequency or absolute number data is unclear.

Reviewer #2:

Remarks to the Author:

In the manuscript "A new molecular classification to drive precision treatment strategies in primary Sjögren's syndrome", the authors performed a comprehensive molecular profiling of patients with pSS by analyzing transcriptomics data with a semi-supervised robust approach and discriminated 4 clusters. Analysis of additional omics (GWAS, methylome), biological features (flow cytometry, serum cytokine expression), and clinical characteristics (PGA, ESSDAI, ESSPRI, and several serological parameters) were assessed to further validate the clusters and establish their functionality. Based on findings above, a composite model was set up to predict the belonging of a patient to one of the 4 clusters. The manuscript is well organized and clearly structured, although several questions are needed to be answered to make it more clinically relevant and valuable.

1. In the manuscript, serum IFN-alpha was not associated with ESSDAI. No significant differences were seen between the 4 clusters as well, although a lower mean score was observed in C2 and the highest score seen in C4. This might be due to the insufficiency of the global ESSDAI score to exhibit the heterogeneity of pSS. I would like to know more about the situation when we look at each domain, which may show a better picture and connections between different molecular signatures and systemic involvements.
2. A previous study published in Lancet Rheumatol, as also mentioned by the authors and listed as Reference No. 41, has proposed a symptom-based stratification method to exploit the heterogeneity of pSS. Four subgroups were identified as Low symptom burden (LSB), High symptom burden (HSB), dryness dominant with fatigue (DDF), and Pain dominant with fatigue (PDF). Of note, IFN signaling pathway was proposed to be the most significant signature in pSS patients, which is consistent to this manuscript. At the level of individual transcriptomic modules, IFN module activity scores were highest in the LSB subgroup, followed by DDF subgroup. However, DDF subgroup had high activity score for mature B-cell modules, which discriminated it from LSB. It appears LSB and DDF subgroup share somewhat similar features with C1 and C3 respectively. However, in this manuscript, no significant difference was seen between the 4 clusters for both the global ESSPRI score and its 3 components. The authors should explain the discrepancy between the 2 stratification methods, and how we can implement the finding into precision treatment strategies the most?

Reviewer #3:

Remarks to the Author:

The authors propose a molecular classification of primary Sjögren's syndrome (pSS) that consists of four subgroups and is primarily developed from transcriptomics data. The subgroups were developed by a previously developed semi-supervised clustering method. The data set consisted of 304 cases and 330 controls and a 75%/25% derivation/validation split was used for development of the clusters.

Face validity and properties of the four clusters were examined using a range of molecular and clinical

data that were not used in deriving the clusters. Examination of 257 top genes discriminating genes showed that three of the clusters were enriched in IFN signaling, lymphoid lineage pathways, and inflammatory & myeloid lineage transcripts respectively. Analysis of other molecular data including GWAS, methylation, peripheral blood counts from flow cytometry, and cytokines were done to identify differences in these data types among the clusters. Further analysis of clinical symptoms and serological characteristics showed differences among the clusters.

To be able to classify a future patient, the authors developed a composite predictive model with xgboost-tree from the genes that were found to discriminate among the clusters based on the Boruta algorithm. A step model was developed where the first model consists of 3 classification trees composed of 10 genes and is used to predict C4 vs. not-C4. If the patient does not belong to C4 cluster, a second model consisting of classification trees composed of 31 genes is applied to predict C1, C2 or C3 clusters. The composite model when applied to the derivation data set performed well with accuracy >95%. The predictive model was also applied to an independent cohort of 37 pSS patients to predict cluster membership.

The data sets, clustering and development of the predictive model are described in adequate detail, and the experimental protocol of using 75/25 derivation/validation split is appropriate. Model performance is primarily evaluated using accuracy and is appropriate. The conclusions from the modeling analyses and data interpretation are valid and reliable.

The following minor suggestions can improve the manuscript:

1. Is there a reason why the predictive model was not evaluated on the validation data set?
2. Consider reporting the area under the Receiver Operating Characteristic curve for the predictive model. Four separate ROC curves can be reported, one for each cluster vs. the rest.

Reviewer #4:

Remarks to the Author:

This is a high quality multi-institutional analysis of SS patients a complex multi-factorial diseases. The authors used RNA seq and other multi-omic data to identify & validate 4 subtypes.

Can the authors elaborate if any previous studies have done any work on subtyping this syndrome using any or a subset of the data types that is collected here? So expression, methylation, SNPs, flow or cytokines.

The methylation analysis appears somewhat arbitrary using the Delta-beta threshold of 0.075, how was this threshold chosen? Is there any data that supports the choice of this threshold, and how dependent are the results on this threshold?

Next on line 271 another threshold is defined at 0.15, any explanation why these two thresholds are needed and how this 2nd analyses differs from the first one?

We would like to thank the associate editor of Nature Communications for considering our manuscript entitled “A new molecular classification to drive precision treatment strategies in primary Sjögren’s syndrome” for publication in Nature Communications and for the opportunity to resubmit a revised version for further consideration.

We are deeply grateful for the considerable efforts on the part of the reviewers to assess the manuscript thoroughly and for their critiques. The positive consideration given by the four reviewers and their thoughtful suggestions are much appreciated.

We have made the necessary changes and addressed the issues raised by the reviewers and would like to highlight in a point-by-point manner the changes made and how raised issues and concerns were addressed.

Reviewer #1

1- The composite predictive model was developed/trained using the smaller inception cohort then tested on the larger cross-sectional cohort. Yet, the approach used to select which genes were to be used in the composite predictive model utilized RNA-Seq data from all of the subjects. Whether this constitutes a true test of the model is thus unclear. The study would be further strengthened by testing the composite predictive model using data from different subjects that were not used to develop the model. In their favor, the authors have facilitated the capacity of others to test the model in their own datasets.

We thank the reviewer for this comment that helps us to understand that our description of the algorithm’s training might be unclear.

We confirm, as well understood by reviewer 3, that we have used patients from the cross-sectional cohort to train the predictive model. This cohort was initially divided into discovery and validation subgroups to respectively determine and confirm the different clusters. The process for the development of the composite model begins with a training phase on the validation subgroup (77 patients). Next, we have evaluated the model on the discovery dataset (227 patients), consisting of patients who were not seen in the training phase. Once we achieved satisfactory precision on this evaluation dataset (>95%) we have run the model on the new inception cohort to see how well the algorithm can perform on patients who were not part of the cross-sectional cohort. Our main objective there was to provide additional evidence that the predictive model didn’t overfit during the training phase and can be used on an external dataset. We understand that reviewer 1 is also concerned by the approach we used to perform our genes selection. Before the training phase of the composite model, we *performed dimensional reduction of our dataset using the patients from the discovery and validation cohorts. As stated in the supplementary materials, the dimensional reduction of our dataset is composed of the following steps: 1-we first identified 1154 genes differentially expressed (DEGs) in the 4 clusters; 2-we used the Boruta algorithm [Kursa, 2010] to select genes that contribute the most to differentiate these clusters; 3-we finally found 255 candidate genes to train the C4 model and 597 candidate genes to train the C1-C2-C3 multiclass prediction model.* This step is part of the data preprocessing framework, reducing the number of variables describing our patients and allowing us to be more robust to overfitting. The genes

selected to represent the patients at this step are not the genes picked by the model at the end of its training but they represent a simplification of our dataset to optimize the model's training process. The 40 genes selected during the training phase of the composite model were selected based only on the patients from the validation cohort (and validated on the discovery cohort). Finally, once the model was fully trained (on the validation cohort) and validated (on the discovery cohort), we tested it on the patients from the inception dataset, that were neither used in the training/validation operation and in the first dimensional reduction operation.

Kursa, M. B. & Rudnicki, W. R. Feature Selection with the BorutaPackage. J Stat Softw. 36, 11 (2010)

2. Cluster 1 is enriched in disease-associated SNPs (n=35). To what extent if any do eQTLs explain the gene expression responsible for delineating this cluster?

We thank the reviewer for this question. In order to respond to the reviewer question, we used the snp-nexus database (<https://www.snp-nexus.org/>) and select all genes nearby the 35 SNPs. Comparing them to the DEGs in C1, only two genes, coding HLA-C and PSMB9, were differentially expressed (FC=1.54 and 1.52 respectively). Then we decided to test the gene coding HLA-C and PSMB9 with the R package MatrixEQTL [Shabaln, 2016]. We focused solely on cis-eQTLs because we lacked substantial power for trans-eQTLs studies. We adjusted for age and sex and the output p-value was fixed at 10^{-5} and the cis distance at 1Mb. Then we found 16 eQTLs associated with this gene (Table below) and none for PSMB9:

gene	snps	pvalue	FDR	beta
HLA-C	rs2247056	5.05E-41	1.15E-38	0.596766125
	rs2734583	6.29E-26	4.78E-24	0.647025982
	rs2523544	1.29E-18	5.88E-17	0.447730609
	rs3094228	5.51E-14	1.80E-12	0.342977871
	rs2394895	7.96E-13	1.81E-11	0.327856995
	rs3132935	9.93E-13	2.06E-11	0.348801633
	rs887468	1.75E-10	1.79E-09	0.275189105
	rs3130473	2.56E-10	2.43E-09	0.281941851
	rs2517576	9.03E-10	7.63E-09	0.302104793
	rs3095151	4.39E-09	3.45E-08	0.301586516
	rs3115663	4.54E-09	3.45E-08	0.291662834
	rs3094112	4.14E-08	2.48E-07	0.269487605
	rs3094122	1.12E-07	6.37E-07	0.256320602
	rs3130467	1.19E-07	6.62E-07	0.235649231

rs3094220	2.39E-07	1.30E-06	0.245553166
rs3130347	1.18E-06	5.62E-06	0.24516001

Shabalin, A.A. Matrix eQTL:Ultra fast eQTL analysis via large matrix operations. *Bioinformatics*, 28, 10, 1074-1082 (2016).

3. Figure 2 is unclear. In particular, the color scheme in Figure 2B is too complex (too many colors). The authors may consider using letters of the alphabet in lieu of colors. The white boxes in Figure 2A are not adequately described. X axes (1-42) in Figure 2A are not adequately described. Annotation of the most important features of Figure 2A with the abbreviations in Figure 2B would be helpful. The Supplementary Figure 5, on the other hand, is clear and easy to interpret.

We agree with the reviewer comment and have inverted Figure 2 and supplementary Figure 5. Therefore, the new Figure 2 corresponds to the previous supplementary Figure 5 and the previous Figure 2 is now the new supplementary Figure 5. We have, as requested by the reviewer, clarified the new supplementary Figure 5 legend. The new legend is as follows:

Supplementary Figure 5: Fingerprint grid plots mapping transcriptome repertoire perturbations across the four pSS clusters. A collection of 16 transcriptome datasets spanning a wide range of immunological and physiological states were used as input to construct a module repertoire (Altman, 2020). Each dataset was independently clustered via k-means clustering. Gene co-clustering events were recorded in a table, where the entries indicate the number of datasets in which co-clustering was observed for a given gene pair. The co-clustering table served as the input to a weighted co-clustering graph, where the nodes represent genes and the edges represent co-clustering events. The largest, most highly weighted sub-networks among a large network were identified mathematically and assigned a module ID. The genes constituting this module were removed from the selection pool and the process was repeated, resulting in the selection of 382 modules constituted by 14,168 transcripts. The modules were arranged onto the grid as follows: the master set of 382 modules was partitioned into 38 clusters (or aggregates) based on similarities among their module activity profiles across the sixteen reference datasets (A1-A38). A subset of 27 aggregates comprising 2 modules or more in turn occupied a line on the grid. The length of each line was adapted to accommodate the number of modules assigned to each cluster. Changes in transcript abundance at the module level were mapped onto this grid and represented by color spots of varying intensity. (a) Changes in blood transcript abundance in subjects from each cluster compared to healthy volunteers (HV) with a fold change cut-off = 1.5 and a FDR adjusted p-value < 0.05 are represented on the fingerprint grid plot. The modules occupy a fixed position on the fingerprint grid plots. An increase in transcript abundance for a given module is represented by a red spot; a decrease in abundance is represented by a blue spot. Modules arranged on a given row belong to a module aggregate (here denoted as A1 to A38). Changes measured at the “aggregate-level” are represented by spots to the left of the grid next to the denomination for the corresponding aggregate. The

colors and intensities of the spots are based on the average across each given row of modules. A module annotation grid (**b**) is provided where a color key indicates the functional associations attributed to some of the modules on the grid. Positions on the annotation grid occupied by modules for which no consensus annotation was attributed are colored white. Positions on the grid for which no modules have been assigned are colored grey.

4. A limitation of the study includes the use of data only from peripheral blood. This is mentioned in the Discussion. This limitation affects interpretation of some of the results. For example, on lines 327-329, the authors seem to suggest that a reduction of peripheral blood pDC is incongruent with the notion that pDC are thought to be the major interferon alpha producers. The authors do not consider published data showing that pDC are enriched in the salivary glands of Sjogren's patients and the possibility that tissue sites may be the major source of interferon alpha in these individuals.

Again, we totally agree with the reviewer concern. We therefore took up the point raised by the reviewer and incorporated it into our discussion. We have then added another reference on pDCs (new reference 45: Hillen et al. Front immunol, 2019).

5. There is considerable overlap of interferon signatures among Clusters 1, 2 and 4. The description of Cluster 4 as “rather of Type II” interferon cluster (Abstract line 61) is an oversimplification, as this cluster also contains strong enrichment of Type 1 interferon modules. Similarly, characterization of Cluster 3 as the B lymphocyte patient cluster may be over-simplified. Other clusters show some enrichment in B cell features, including IPA B cell signaling pathways (also enriched in Clusters 1 and 4, Suppl Table 2), hypergammaglobulinemia (high in Clusters 1 and 3), elevated RF (also observed in Cluster 1), circulating free light chains (elevated in all clusters vs. HV), SSA/SSB positivity (prevalent in Clusters 1 and 3), ESSDAI biological domain (elevated in Clusters 1 and 3). Over-simplifying description of the patient clusters could unintentionally create confusion in the field.

We agree with the reviewer that the clusters’ description we proposed as a summary of their complete pathway analysis is over-simplified and may impair the content of our messages. In this work, we have tried to highlight the molecular features that can best characterized each independent cluster knowing that there is a continuum in the molecular abnormalities observed in pSS. Hence, we have modified the text to more precisely reflect the different facets that characterize our clusters. Of note, the description of the clusters was transferred from the abstract to the “introduction” section when reducing the abstract length in line with the Nature publishing rules. The clusters are now described as follows in the “introduction” and in the “discussion” sections.

“The Cluster 1 (C1), C3 and C4 display a high interferon (IFN) signature reflecting the pathological involvement of the IFN pathway, but with various Type I and II IFN gene enrichment. C1 has the strongest IFN signature with both Type I and Type II gene enrichment when compared to C3 (intermediate) and C4 (lower). C4 has a Type II gene enrichment

stronger than Type I and equivalent to C3 while C3 has the opposite composition. C2 exhibits a weak Type I and Type II IFN signature with no other obvious distinguishable profile relative to HV. We further characterized C1, C3 and C4 using multi-omics and clinical data. C1 patients present a high prevalence of SNPs, C3 patients an involvement of B cell component more prominent than in the other clusters and especially an increased frequency of B cells in the blood while C4 patients have an inflammatory signature driven by monocytes and neutrophils, together with an aberrant methylation status”

6. Provision of complete lists of differentially expressed genes would be desirable to enhance accessibility of the study results.

We thank the reviewer for having raised this point. Complete lists instead of top 100 of differentially expressed genes have been added in supplementary Table 1.

7. Mention of 2016 ACR/EULAR criteria in the opening sentence of the Results section is confusing, as the Methods state that AECG criteria were used for this study.

We thank the reviewer for having raised this point. The AECG criteria were indeed used to include patients in the study. The text has been modified accordingly.

8. A clear definition of disease duration used in Table 1 is needed. Is this from date of diagnosis or date of first symptom(s), as both fields of data are listed in the Supplementary Methods?

The disease duration has been calculated as the time between the date of the first symptoms (or date of disease onset) and the date of the visit where all data used in this work were collected.

The clinical data section of the Supplementary Materials was modified to confirm that the date of disease onset was only recorded. The date of diagnosis was not considered relevant for the purpose of this work as the diagnosis may be done significantly after the first molecular disturbances occur.

9. Inclusion of Healthy Volunteer data in Figure 6C would be helpful.

We thank the reviewer for having raised this point. The Figure 6C shows results from a one-way ANOVA between clusters and HV. The significance between the cluster and HV is represented by a p-value and the direction of the association is shown as the z-score. We have modified the legend accordingly.

10. Figure 5B colors denoting 5'UTR and IGR and difficult to distinguish.

We thank the reviewer for having raised this point. The color denoting IGR has been modified in Figure 5B.

11. In Suppl Figure 6, the significance designations in the figures are too small to be legible. These should be enlarged or the data presented in tables.

We thank the reviewer for having raised this point. We have enlarged the figure.

12. In Suppl Figure 13, whether the flow cytometry items utilize frequency or absolute number data is unclear.

We thank the reviewer for having raised this point. The flow cytometry items utilize frequency. We have modified the legend accordingly.

Reviewer #2

1. In the manuscript, serum IFN-alpha was not associated with ESSDAI. No significant differences were seen between the 4 clusters as well, although a lower mean score was observed in C2 and the highest score seen in C4. This might be due to the insufficiency of the global ESSDAI score to exhibit the heterogeneity of pSS. I would like to know more about the situation when we look at each domain, which may show a better picture and connections between different molecular signatures and systemic involvements.

We thank the reviewer for this comment. We confirm that we have explored this aspect.

Although the serum IFN-alpha was not associated with global ESSDAI, we analyzed its association with each domain of ESSDAI (*Glandular, Articular, Cutaneous, Respiratory, Renal, Muscular, Peripheral nervous, Central nervous, Hematological and Biological*).

A cut-off has been fixed at the LOD (correspond to mean (HV) + 2SD (HV)) to characterize 42 patients (54%) with high levels of IFN-alpha (hi-IFN α) and 36 patients (46%) with low levels of IFN-alpha (lo-IFN α).

Through a first univariate analysis (Fisher-exact test) we identified an association of the IFN α levels with the hematological ($p=0.004$) and the biological ($p=0.004$) domains. Actually, there was an over-representation of patients with a positive hematological domain (26%) and patients with a positive biological domain (52%) in the hi-IFN α population. All results are present in the Table below. These findings are reflecting the clusters' characteristics observed when analyzing independently the different items of the hematological and biological domains of ESSDAI. Hyperglobulinemia, high levels of antinuclear antibodies, reduced C4 and

lymphopenia characterized C1 patients and in a lesser extent C3 patients who presented the higher IFN signature (Figure 6 and Table 2).

The second multivariate analysis (PLS-DA) did not allow to extract other association between IFN-alpha and ESSDAI domains.

This point raised by the reviewer has been added in the “Results” section.

		hi-IFN α (n = 42)	lo-IFN α (n = 36)	p-value
Glandular	n (%)	9 (21)	5 (14)	0,555
Articular	n (%)	14 (33)	12 (33)	1
Cutaneous	n (%)	3 (7.1)	0 (0)	0,245
Respiratory	n (%)	4 (9.5)	4 (11)	1
Renal	n (%)	2 (4.8)	0 (0)	0,496
Muscular	n (%)	0 (0)	0 (0)	-
Peripheral nervous	n (%)	3 (7.1)	4 (11)	0,697
Central nervous system	n (%)	1 (2.4)	1 (2.8)	1
Hematological	n (%)	11 (26)	1 (2.8)	0,004
Biological	n (%)	22 (52)	7 (19)	0,004

n: number of patients with available information

Statistical tests performed: fisher-exact test of independence for categorial variable

2. A previous study published in Lancet Rheumatol, as also mentioned by the authors and listed as Reference No. 41, has proposed a symptom-based stratification method to exploit the heterogeneity of pSS. Four subgroups were identified as Low symptom burden (LSB), High symptom burden (HSB), dryness dominant with fatigue (DDF), and Pain dominant with fatigue (PDF). Of note, IFN signaling pathway was proposed to be the most significant signature in pSS patients, which is consistent to this manuscript. At the level of individual transcriptomic modules, IFN module activity scores were highest in the LSB subgroup, followed by DDF subgroup. However, DDF subgroup had high activity score for mature B-cell modules, which discriminated it from LSB. It appears LSB and DDF subgroup share somewhat similar features with C1 and C3 respectively. However, in this manuscript, no significant difference was seen between the 4 clusters for both the global ESSPRI score and its 3 components. The authors should explain the discrepancy between the 2 stratification methods, and how we can implement the finding into precision treatment strategies the most?

In pSS, previous studies from the same group than Ref 41 (now Ref 42 in the revised version of the manuscript) have also shown that higher pro-inflammatory cytokine levels (IP-10, TNF α , IFN α , IFN- γ , and LT- α) were associated with lower patient-reported fatigue (Davies, 2019) and that therapeutic effects of RSV-132 (a new RNAses compound) resulted in a significant improvement of fatigue correlated with the increased expression of IFN-stimulated gene (Posada, 2020). However, analysis of IFN-stimulated gene expression did not find any correlation between the IFN modular score and fatigue in another pSS dataset

(Seguier, 2020). Finally, in another study, pSS patients with an IFN signature reported a better quality of life than those without (Bodewes, 2020). Importantly, in the last study, patients treated with hydroxychloroquine displayed a decreased IFN signature with no benefit on fatigue (Bodewes, 2020). Therefore, the observations are often discordant.

We agree with the reviewer that there are some similarities between the clusters described by Tarn et al (ref 41) and our clusters concerning the IFN signature. Thus, C1 is close to the LSB cluster, C2 to the PDF cluster, C3 to the DDF cluster and C4 to the HSB cluster as described in the figure below.

Chaussabel module activity scores adjusting for batch differences for the transcriptomics datasets centred on the mean for each module as described by Tarn et al (2019) (a) and obtained in our study (b). Shown are the top 3 modules with significant differences between subgroups. Negative values imply inhibition and positive values imply activation. The error bars represent 95% CIs. IFN=interferon

In order to respond to the reviewer question, we grouped our patients according to the symptom-based classification proposed in reference 41. However, we do not have, in our cohort, data concerning depression and anxiety. Therefore, the stratification is biased and is based only on data from the ESSPRI regarding dryness, fatigue and pain.

Characteristics	N	C1, N = 56 ¹	C2, N = 43 ¹	C3, N = 30 ¹	C4, N = 21 ¹	p-value ²
DDF	150	22 (39%)	12 (28%)	13 (43%)	9 (43%)	0.5
HSB	150	14 (25%)	12 (28%)	10 (33%)	3 (14%)	0.5
LSB	150	17 (30%)	10 (23%)	6 (20%)	3 (14%)	0.5
PDF	150	3 (5.4%)	9 (21%)	1 (3.3%)	6 (29%)	0.005
Characteristics	N	DDF, N = 56 ¹	HSB, N = 39 ¹	LSB, N = 36 ¹	PDF, N = 19 ¹	p-value ²
C1	150	22 (39%)	14 (36%)	17 (47%)	3 (16%)	0.14
C2	150	12 (21%)	12 (31%)	10 (28%)	9 (47%)	0.2
C3	150	13 (23%)	10 (26%)	6 (17%)	1 (5.3%)	0.3
C4	150	9 (16%)	3 (7.7%)	3 (8.3%)	6 (32%)	0.083

¹ Statistics presented: n (%)

² Statistical tests performed: chi-square test of independence; Fisher's exact test

We were not able to find any enrichment of one of our four clusters in any of the subgroups proposed in reference 41.

However, the positive and quite unique aspect of our approach is that it will allow the Newcastle team who published reference 41 to answer the reviewer's question because we are proposing, for the first time, the means of predicting the membership of patients from this cohort to one of the 4 clusters that we have identified. It is therefore impossible at this stage to say that there are discrepancies between our study and the one published in Lancet Rheumatol. Our group is actually involved in an IMI2 European project called NECESSITY aiming at identifying new response criteria to treatment in the context of pSS. We will then have access to many different cohorts comprising ASSESS and UKPSSR used in the reference 41 paper and expect to have the opportunity to do such analysis.

Davies K, Mirza K, Tarn J, Howard-Tripp N, Bowman SJ, Lendrem D; UK Primary Sjögren's Syndrome Registry, Ng WF. Fatigue in primary Sjögren's syndrome (pSS) is associated with lower levels of proinflammatory cytokines: a validation study. *Rheumatol Int.* 2019 Nov;39(11):1867-1873.

Posada J, Valadkhan S, Burge D, Davies K, Tarn J, Casement J, Jobling K, Gallagher P, Wilson D, Barone F, Fisher BA, Ng WF. Improvement of Severe Fatigue Following Nuclease Therapy in Patients With Primary Sjögren's Syndrome: A Randomized Clinical Trial. *Arthritis Rheumatol.* 2020 Aug 15;73(1):143–50.

Seguier J, Jouve E, Bobot M, et al. Paradoxical association between blood modular interferon signatures and quality of life in patients with systemic lupus erythematosus. *Rheumatology (Oxford).* 2020;59(8):1975-1983.

Bodewes ILA, Gottenberg JE, van Helden-Meeuwsen CG, Mariette X, Versnel MA. Hydroxychloroquine treatment downregulates systemic interferon activation in primary Sjögren's syndrome in the JOQUER randomized trial. *Rheumatology (Oxford).* 2020;59(1):107-111.

Reviewer #3

The data sets, clustering and development of the predictive model are described in adequate detail, and the experimental protocol of using 75/25 derivation/validation split is appropriate. Model performance is primarily evaluated using accuracy and is appropriate. The conclusions from the modeling analyses and data interpretation are valid and reliable.

We thank the reviewer for his/her comment on our methodology procedure.

1. Is there a reason why the predictive model was not evaluated on the validation data set?

In fact, we have also evaluated the predictive model in the validation dataset where we achieved 92% of accuracy. However, we have focused, in the text the evaluation of the predictive composite model on the discovery dataset (95%) which is composed of a higher number of patients.

2. Consider reporting the area under the Receiver Operating Characteristic curve for the predictive model. Four separate ROC curves can be reported, one for each cluster vs. the

rest.

We thank the reviewer for the proposal. Adding four separate ROC curves would serve the description of the model's performance. Unfortunately, due to the nature of our model, only the ROC curve for C4 could be accurately reported.

Performance evaluation of the 2-steps composite model with Receiver Operating Characteristic (ROC) curves. a) The ROC curve for the C4 classification part of the composite model. When an observation is not classified as C4 then it is submitted to the C1-C2-C3 classification part of the composite model. b) The ROC curves for each of the clusters predicted in this last part. Patients from the cross-sectional cohort were used to compute these ROC curves : 304 patients were used to compute the C4 AUC curve (36 were predicted as C4) and then 268 patients were used to compute the ROC curves for C1, C2 and C3.

Indeed, in a first step, the composite model assesses the probability that a patient belongs to C4. Then, in a second step, and only if probability of belonging is small, the second part of the model is launched and the probabilities of belonging for C1, C2 and C3 are calculated assuming that the probability for C4 belonging is 0, which is not the case at the first step. Therefore, for C1, C2 and C3, any probability value of belonging to one specific group versus all other would be an approximation. We fear that this approximation could mislead the reader in understanding the composite model and this is why we have preferred a representation of the performance of the algorithm by using a confusion matrix.

Reviewer #4

1- Can the authors elaborate if any previous studies have done any work on subtyping this

syndrome using any or a subset of the data types that is collected here? So expression, methylation, SNPs, flow or cytokines.

To our knowledge, we report herein on the largest ever molecular profiling study carried out in primary Sjögren's syndrome patients and compared to healthy volunteers, using high-throughput multi-omics data (genetic, epigenomic, transcriptomic, combined with flow cytometric data, multiplexed cytokines, as well as classical serology).

There were some publications reporting on patient stratification attempts. Most of them were based on gene expression data with a focus on the IFN pathway involvement. The team of Bodewes et al. (Ref 1) described patient subgroups according to their IFN pathway involvement with specific attention to Type I and Type II gene enrichments. James et al. (Ref 2) found three clusters using gene expression microarray characterized by different levels of IFN and inflammation. Davies et al. (Ref 3) proposed a stratification based on absolute cell counts and lastly Tarn et al. (Ref 4) described patient clinical phenotypes characterized *a posteriori* at the molecular level with gene expression data. These works provide good basis for building a molecular taxonomy of Sjogren disease. Our integrative approach using multi-omics and patient clinical characteristics allows going further in understanding the Sjogren disease heterogeneity.

We have described shortly these studies in the “discussion” part and added the corresponding references.

1-Bodewes, I. L. A. et al. Systemic interferon type I and type II signatures in primary Sjögren's syndrome reveal differences in biological disease activity. *Rheumatology (Oxford)* 57, 921-930 (2018).

2-James, J. A. et al. Unique Sjögren's syndrome patient subsets defined by molecular features. *Rheumatology (Oxford)* 59, 860-868 (2020).

3-Davies R et al. Patients with Primary Sjögren's Syndrome Have Alterations in Absolute Quantities of Specific Peripheral Leucocyte Populations. *Scand J Immunol.* 86, 491-502 (2017).

4-Tarn, J. R. et al. Symptom-based stratification of patients with primary Sjögren's syndrome: multi-dimensional characterisation of international observational cohorts and reanalyses of randomised clinical trials. *Lancet Rheumatol.*1, e85–94 (2019).

2- The methylation analysis appears somewhat arbitrary using the Delta-beta threshold of 0.075, how was this threshold chosen? Is there any data that supports the choice of this threshold, and how dependent are the results on this threshold?

Next on line 271 another threshold is defined at 0.15, any explanation why these two thresholds are needed and how this 2nd analyses differ from the first one?

β values constitute the ratio of all methylated probe intensities over the total signal intensities (methylated and unmethylated) and have a range between 0 and 1. β values are an approximation of the percentage of methylation (0-100%). Many $\Delta\beta$ thresholds are described in the literature and the most frequently used for whole blood studies in autoimmune diseases are 0.05 (5% difference) and 0.1 (10% difference). However, extreme values of 0.03 and 0.4 were also reported (Ref 1 et 2).

For example, Braekke Norheim et al. (2016) studied the methylation status in whole blood of 48 pSS patients with high (n=24) and low (n=24) degree of fatigue. They choose a false discovery rate–corrected $P < 0.05$ and a mean difference in DNA methylation level between the two patient groups of at least 3% (ΔBeta threshold of 0.03) (Ref 1). Imgenberg-Kreuz et al (2016) analysed whole blood from 100 pSS patients and 400 healthy volunteers (HV). They fixed a Bonferroni adjusted p-value of $1.3 \cdot 10^{-7}$ in a first step and in a second step, they focused the DMPs with an absolute ΔBeta of 0.1 (Ref 3). In 2018, the same group (Ref 4) compared whole blood methylation between 548 SLE patients and 587 HV. They applied a Bonferroni corrected p-value of $1.3 \cdot 10^{-7}$ and an absolute ΔBeta threshold of 0.05. Yeung et al. (2017), compared whole blood methylation between 12 SLE patients and 10 healthy controls They applied an adjusted p-value of 0.05 and an absolute ΔBeta threshold of 0.1 (Ref 5). Finally, Turell et al. (2020) analysed methylation in whole blood between 189 pSS patients and 220 HV. They applied 2 steps of analysis. In the first step, they applied only a Bonferoni corrected threshold of p-value $< 6.4 \cdot 10^{-8}$. In the second step, they described the top 10 DMPs with an absolute $\Delta\text{Beta} > 0.1$ and founded genes associated with IFN signature (Ref 6).

In view of what was described in the literature, we applied a ΔBeta threshold of 0.075. We assume that a ΔBeta threshold of 0.05 is too low and that we could have missed DMPs when applying a ΔBeta threshold of 0.1 in first intention. To illustrate this, we have tested thresholds of 0.05, 0.75, 0.1 and 0.15. The table below represents the numbers of DMPs and genes obtained with these different thresholds:

		C1	C2	C3	C4
abs(ΔBeta) >0.05	DMPs	1508	7	759	17150
	Genes	918	6	482	6256
abs(ΔBeta) >0.075	DMPs	145	2	96	8445
	Genes	87	2	56	4711
abs(ΔBeta) >0.1	DMPs	37	0	38	4575
	Genes	24	0	26	2270
abs(ΔBeta) >0.15	DMPs	13	0	17	1194
	Genes	10	0	11	761

If we had applied a ΔBeta threshold of 0.05, we would have obtained 17150 DMPs corresponding to 6256 genes in C4 and it would have been very difficult to interpret the signification of these defects in methylation. If we had applied a ΔBeta threshold of 0.1 in a first step, we could not have observed the defect in methylation of C2 which is lower than in the other clusters. However, the reviewer's comment is relevant because when applying a ΔBeta threshold of 0.05, we move in C2 from 2 (with a ΔBeta threshold of 0.075) to 7 DMPs.

These 7 DMPs were also present in the three other clusters and corresponded to 5 genes involved in the IFN signature (IFIT1, IFITM1, NLRC5, IFI44L and MX1 that presents two DMPs). With a ΔBeta threshold of 0.075, only NLRC5 and MX1 were found. Consequently, the ΔBeta threshold of 0.05 reinforces the message already described in our manuscript that the methylation anomaly of genes associated with the IFN pathway is also present in C2.

The ΔBeta cut-off at 0.075 identified 8445 DMPs corresponding to 4711 hypo- and hyper-methylated genes in C4, showing an aberrant methylation status of this cluster. The ΔBeta cut-off at 0.15 was basically used in a second step to identify the most robust and significant signature of hypo- and hyper-methylated genes in C4. Performing the analysis on so many DMPs would have led to a less relevant analysis of the genes involved. Finally, this 2-step analysis is similar to Imgenberg-Kreuz et al. (2018) (**Ref 3**) and permits to conclude that the default of methylation in C4 concerned mainly the neutrophil degranulation pathway which is a specificity of this cluster.

- 1- Brække Norheim, K et al. Epigenome-wide DNA methylation patterns associated with fatigue in primary Sjögren's syndrome. *Rheumatology*, 55, 6, 1074-1082 (2016).
- 2- Haider, Z et al. An integrated transcriptome analysis in T-cell acute lymphoblastic leukemia links DNA methylation subgroups to dysregulated TAL1 and ANTP homeobox gene expression. *Cancer Medicine*, 8, 1: 311-324 (2019).
- 3- Imgenberg-Kreuz, J et al. Genome-wide DNA methylation analysis in multiple tissues in primary Sjögren's syndrome reveals regulatory effects at interferon-induced genes. *Annals of the Rheumatic Diseases*, 75, 11, 2029–2036 (2016).
- 4- Imgenberg-Kreuz, J et al. DNA methylation mapping identifies gene regulatory effects in patients with systemic lupus erythematosus. *Annals of the Rheumatic Diseases*, 77, 5, 736–743 (2018).
- 5- Yeung KS, et al. GenomeWide DNA Methylation Analysis of Chinese Patients with Systemic Lupus Erythematosus Identified Hypomethylation in Genes Related to the Type I Interferon Pathway. *PLoS ONE*, 12, 1, e0169553 (2017).
- 6- Teruel, M et al. An integrative multi-omics approach in Sjögren's Syndrome identifies novel genetic drivers with regulatory function and disease-specificity. *medRxiv* (2020)

Reviewers' Comments:

Reviewer #1:

Remarks to the Author:

The clarity of the figures and supplemental figures is improved. Clusters are more accurately summarized. Additional supplemental data are provided. All other concerns have been well-addressed.

Reviewer #2:

Remarks to the Author:

This is a high-quality molecular profiling study carried out in pSS. The issues raised in my review have been clearly addressed in the revision.

The authors have analyzed the association between serum IFN-alpha and each domain of ESSDAI and have tried to group their patients according to the symptom-based stratification model proposed in a previous study. I am glad to know that the authors' group is involved in the IMI2 European project and definitely would like to know more about the performance of this predictive model in other cohorts in the future.

Reviewer #3:

Remarks to the Author:

The authors have revised the manuscript and made several clarifications. I am satisfied with the revisions and have no further suggestions.

Reviewer #4:

Remarks to the Author:

The authors have answered all my comments. I would recommend one thing to the authors, namely to add the response to my question #2 regarding the methylation thresholds as a supplementary note to the manuscript as it strengthens the message.

We would like to thank the associate editor of Nature Communications for considering our manuscript entitled “A new molecular classification to drive precision treatment strategies in primary Sjögren’s syndrome” for publication in Nature Communications and for the opportunity to resubmit a revised version for further consideration.

We have made the necessary changes and addressed the issues raised by the reviewers and would like to highlight in a point-by-point manner the changes made and how raised issues and concerns were addressed.

REVIEWERS' COMMENTS

Reviewer #1 (Remarks to the Author):

The clarity of the figures and supplemental figures is improved. Clusters are more accurately summarized. Additional supplemental data are provided. All other concerns have been well-addressed.

We thank the reviewer for his comments which improved the quality of the manuscript.

Reviewer #2 (Remarks to the Author):

This is a high-quality molecular profiling study carried out in pSS. The issues raised in my review have been clearly addressed in the revision.

The authors have analyzed the association between serum IFN-alpha and each domain of ESSDAI and have tried to group their patients according to the symptom-based stratification model proposed in a previous study. I am glad to know that the authors' group is involved in the IMI2 European project and definitely would like to know more about the performance of this predictive model in other cohorts in the future.

We thank the reviewer for his comments which improved the quality of the manuscript.

Reviewer #3 (Remarks to the Author):

The authors have revised the manuscript and made several clarifications. I am satisfied with the revisions and have no further suggestions.

We thank the reviewer for his comments which improved the quality of the manuscript.

Reviewer #4 (Remarks to the Author):

The authors have answered all my comments. I would recommend one thing to the authors, namely to add the response to my question #2 regarding the methylation thresholds as a supplementary note to the manuscript as it strengthens the message.

We thank the reviewer for his comments which improved the quality of the manuscript. The response to the reviewer previous question#2 has now been added to the manuscript both in the Methods and Results sections (Supplementary Figure 6).